# Topology of Reasoning: Understanding Large Reasoning Models through Reasoning Graph Properties

**Gouki Minegishi**[1]  **Hiroki Furuta**[2†]  **Takeshi Kojima**[1]  **Yusuke Iwasawa**[1]  **Yutaka Matsuo**[1]
[1]The University of Tokyo, [2]Google DeepMind
{minegishi,furuta,t.kojima,iwasawa,matsuo}@weblab.t.u-tokyo.ac.jp

## Abstract

Recent large-scale reasoning models have achieved state-of-the-art performance on challenging mathematical benchmarks, yet the internal mechanisms underlying their success remain poorly understood. In this work, we introduce the notion of a reasoning graph, extracted by clustering hidden-state representations at each reasoning step, and systematically analyze three key graph-theoretic properties: cyclicity, diameter, and small-world index, across multiple tasks (GSM8K, MATH500, AIME 2024). Our findings reveal that distilled reasoning models (e.g., DeepSeek-R1-Distill-Qwen-32B) exhibit significantly more recurrent cycles (about 5 per sample), substantially larger graph diameters, and pronounced small-world characteristics (about 6x) compared to their base counterparts. Notably, these structural advantages grow with task difficulty and model capacity, with cycle detection peaking at the 14B scale and exploration diameter maximized in the 32B variant, correlating positively with accuracy. Furthermore, we show that supervised fine-tuning on an improved dataset systematically expands reasoning graph diameters in tandem with performance gains, offering concrete guidelines for dataset design aimed at boosting reasoning capabilities. By bridging theoretical insights into reasoning graph structures with practical recommendations for data construction, our work advances both the interpretability and the efficacy of large reasoning models. Implementation available here: https://github.com/gouki510/Topology_of_Reasoning

## 1 Introduction

Recent advances in large reasoning models, such as OpenAI-o1 families [47], extended thinking mode in Gemini [29], Claude [2], Grok [66], and DeepSeek-R1 [10], have achieved striking performance gains pushing the frontier across expert-level coding, competitive math, and PhD-level science questions. These recent reasoning models are characterized with to think and reason for longer before responding. Inspired by the breakthroughs in reasoning capabilities, novel methods to imitate reasoning abilities with smaller models have been developed such as supervised fine-tuning (SFT) techniques [44, 69] and distillation [10]. However, despite these notable successes, the internal mechanisms enabling their remarkable reasoning capabilities remain unclear, particularly in comparison to traditional, non-reasoning models.

To understand the key factors behind the success of recent reasoning models, we introduce the concept of a *reasoning graph* (Figure 1). In mathematical tasks, for example, a reasoning graph can be defined as the path through simple computational states (e.g., addition or subtraction) toward the final answer, where each state corresponds to a node in the graph. Prior to recent breakthroughs in reasoning models, some previous works [48, 63] have empirically and theoretically shown that LLMs

---

†Work done as an advisory role only.

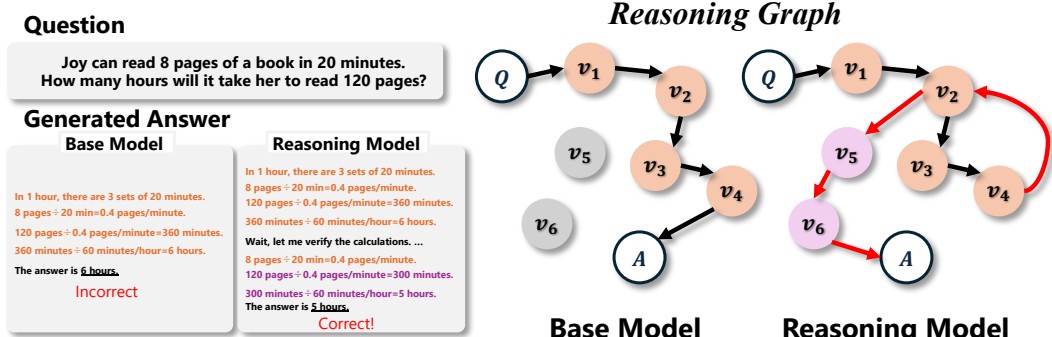

Figure 1: Illustration of the concept of *reasoning graphs*, comparing base models and large reasoning models. Nodes represent simple computational states (e.g., calculation steps shown on the left), with paths leading to the final answer constituting the reasoning graph. We analyze graph-theoretic properties of reasoning graphs, including *cyclic structures*, *diameter*, and *small-world* characteristics. Examining these structural distinctions enables us to better understand and recent performance improvements in challenging mathematical tasks.

employing chain-of-thought prompting achieve higher accuracy by traversing this graph step-by-step. In this work, we analyze the reasoning graphs of large reasoning models from a graph-theoretic perspective, aiming to identify unique structural properties that contribute recent breakthroughs in reasoning performance.

First, we extract reasoning graph nodes by clustering hidden states of LLMs using kmeans during reasoning tasks. Then, for each reasoning task sample, we construct a reasoning graph by connecting the nodes visited by the model during inference and analyze its properties. Visualizing the reasoning graph with t-SNE clearly demonstrates that reasoning graphs from large reasoning models include *cycles* and have a broader exploration range compared to base models. Quantitatively assessing these cyclic properties of reasoning graph confirms that large reasoning models exhibit significantly more cycles than base models. Furthermore, the proportion of reasoning graphs containing cycles increases progressively with task difficulty, as observed across datasets GSM8K [7] (easier), MATH500 [23] (intermediate), and AIME 2024 [11] (more challenging). To quantitatively examine whether large reasoning models explore a broader range of nodes during inference, we compared the *diameters* of their reasoning graphs. We observed notably larger diameters for large reasoning models, suggesting that they explore a wider variety of reasoning states, potentially enabling more sophisticated inference strategies. To gain deeper insights into the structural underpinnings of reasoning ability, we assess the *small-world* characteristics of these graphs, revealing that large reasoning models construct graphs exhibit significantly higher small-world properties. This small-world structures indicate that reasoning graph of large reasoning model have dense local clustering structures which likely contribute to improved reasoning performance. These graph-theoretic properties become more pronounced with increasing model size, suggesting that greater model capacity facilitates the formation of graph structures beneficial for reasoning. These distinct graph-theoretic characteristics provide critical insights into the mechanisms underlying enhanced reasoning performance.

Furthermore, to connect insights derived from reasoning graph properties directly to practical LLM training, we examine reasoning graphs through the lens of FT, particularly analyzing the s1 method [44]. Training the base model using the s1 dataset leads to increased graph diameters, and notably, training on the improved s1-v1.1 dataset, which achieves higher performance, results in even larger graph diameters. These results indicate that more effective SFT data for reasoning can be characterized by distinctive graph properties, suggesting practical guidance for designing better data construction methods.

In summary, our contributions are below:

- **Graph-Theoretic Analysis of Enhanced Reasoning** (Section 4): Through empirical analyses across multiple datasets, we identify distinctive graph properties of large reasoning models, including (1) increased cyclic behavior, (2) larger graph diameters, and (3) heightened small-world characteristics. These structural patterns offer key insights into the mechanisms behind recent breakthroughs in the reasoning performance of LLMs, highlighting how advanced models explore a broader range of reasoning states and transition between them more effectively.

- **Insights for Effective SFT Data Construction** (Section 5): By examining how SFT datasets influence reasoning graph characteristics, we reveal that refined datasets lead to larger graph diameters and better reasoning performance. These insights provide actionable guidelines for designing training data to explicitly enhance reasoning capabilities.

## 2 Related Works

**Approaches to Enhanced Reasoning**  Recent methods for enhancing reasoning in language models include: (1) search-based methods that utilize additional inference-time computation or iterative self-improvement, and (2) reinforcement learning (RL)-based fine-tuning, which has notably driven significant performance breakthroughs. Search-based methods involve external computation of inference time such as parallel sampling [34, 5], sophisticated verifier-based searches [38, 70, 62], or internal in-context refinement and self-correction strategies [17, 18, 71, 33, 27]. Despite their effectiveness, these methods often require careful design or redundant computation. RL methods autonomously discover effective reasoning strategies, encompassing both off-policy [74, 24] and on-policy techniques [75, 30, 9]. Recent advances like DeepSeek-R1 [10], leveraging algorithms such as PPO [50] and GRPO [51], demonstrate significant improvements through structured reasoning traces [35, 73]. A notable phenomenon is the emergence of the *"aha moment"*, characterized by models spontaneously revising their reasoning strategies, inspiring new training approaches such as s1 [44] and Think-DPO [69].

Motivated by these findings, this research elucidates reasoning improvements by analyzing underlying *reasoning graph* structures.

**Analytical Studies on Reasoning Capabilities**  Earlier studies prior to recent breakthroughs in RL-based reasoning emphasized reasoning graphs to explain LLM capabilities. Previous work [48] theoretically and empirically demonstrated that reasoning capabilities emerge due to the locality property inherent in natural language data; specifically, models achieve better accuracy by traversing intermediate variables frequently co-occurring during training. Additionally, Wang et al. [63] proposed extracting reasoning graphs by clustering internal model states using $K$-means, hypothesizing that pre-training data enables a random walk over these graph nodes. Other studies [13, 67, 6, 59] have created simple toy tasks with explicit reasoning graphs to better understand the mechanisms underlying reasoning abilities in language models. More recently, DeepSeek-R1 [10] have demonstrated significant improvements through RL without explicit reasoning supervision. Analyses following this advancement have explored steering vectors [58], cognitive behaviors [19], and anthropomorphic expressions [68].

However, while recent reasoning models are featured with enhanced reasoning traces, no studies have analyzed them through reasoning graph perspective. Addressing this gap is essential for understanding current advancements.

## 3 Graph Properties of LLM's Reasoning Process

### 3.1 Extracting Reasoning Graph from LLM's Representations

Let $\mathcal{D} = \{x_n\}_{n=1}^N$ be our evaluation set of $N$ questions. For each question $x \in \mathcal{D}$, we prompt the model to generate a sequence of intermediate reasoning steps $R = (r_1, r_2, \ldots, r_T)$, where each $r_t$ is delimited by a newline character ('\n') and thus represents one reasoning step. We denote the total number of reasoning steps per question as $T$, and the token length of a segment as $L_t$. Let $h_{t,\mu}^\ell \in \mathbb{R}^d$ be the hidden state at Transformer layer $\ell$ for the $\mu$-th token of segment $r_t$ (illustrated in Figure 2-(a)). and define the segment representation as the mean: $s_t^\ell = \frac{1}{L_t} \sum_{\mu=1}^{L_t} h_{t,\mu}^\ell$.

**Node Definition**  Following the previous work [63], we aggregate all segment representations $\mathcal{S} = \left\{ s_t^\ell \mid 1 \leq t \leq T, \ x \in \mathcal{D} \right\}$ and run $K$-means (default $K = 200$) to obtain clusters $\{C_k\}_{k=1}^K$ with centroids $\{c_k\}$. Each centroid $c_k$ corresponds to a node $v_k$ in the reasoning graph:

$$V = \{v_1, \ldots, v_K\}, \quad d(v_i, v_j) = \|c_i - c_j\|_2.$$

The distance $d(v_i, v_j)$ between nodes $v_i$ and $v_j$ is defined as the Euclidean distance between their corresponding centroids $c_i$ and $c_j$. Figure 2-(b) presents representative nodes obtained from clustering

| Node | Generated Reasoning Step |
|------|--------------------------|
| **Multiplicative** (Node 4) | First leg: $10 \times 30 = 300$. 
 Now multiply that by $120 \times 1.157625$. |
| **Additive** (Node 21) | Total chairs: $22 + 66 = 88$. 
 Total sofas: $20 + 64 = 84$. |
| **Wait** (Node 15) | Wait, but we need to add this ... 
 Wait, no. Let me clarify ... |

(a) Extract reasoning graphs from LLMs      (b) Representative Nodes from Reasoning Graph

Figure 2: **(a)** Illustration of the methodology used to extract reasoning graphs from LLMs. **(b)** Representative nodes obtained from clustering the DeepSeek-R1-Distill-Qwen-32B using GSM8K dataset.

the DeepSeek-R1-Distill-Qwen-32B [10] using GSM8K dataset [7]. Each clustered node corresponds to simple computations encountered within tasks. As characteristic of reasoning models, some nodes include the term "wait", indicative of an *"aha moment"* where the model rechecks its outputs. A more fine-grained analysis, leveraging an LLM-as-judge to characterize the semantics of these clusters, is provided in Appendix B.

**Edge Construction**    Informally, the edges in the reasoning graph represent the sequential path of nodes visited by the model for each question during inference. Formally, for each question $x$, let $\pi = (i_1, i_2, \ldots, i_T)$, where $i_t = \arg\min_k \|s_t^\ell - c_k\|_2$ assigns segment $r_t$ to its nearest centroid. We then define the directed-edge set

$$E = \big\{ (v_{i_t} \rightarrow v_{i_{t+1}}) \mid t = 1, \ldots, T-1 \big\},$$

yielding the *reasoning graph* $G = (V, E)$ for that question. The reasoning graph properties (cycle density, diameter, small-world index) are then computed over $G$.

## 3.2 Measuring Graph Properties

Having extracted reasoning graphs from LLM representations, we evaluate their structural properties from three perspectives: (1) Cycles, (2) Diameter, and (3) Small-World index. Simple implementations of each method are provided in Appendix C.

**Cycles**    We detect cycles in the reasoning graph, defined as repeated visits to the same node, excluding self-loops and adjacent duplicates. This is because repetitive behaviors frequently observed in large reasoning models do not represent meaningful cycles [68]. We define the *cycle detection ratio* as the proportion of reasoning graphs containing at least one cycle across all samples. Additionally, we measure the *cycle count* of each reasoning graph as the maximum number of repeated visits to any single node (excluding self-loops, which repeat the same sentence).

**Diameter**    To compute the reasoning graph diameter, defined as the maximum shortest path distance between any two reachable nodes, we run Dijkstra's algorithm [12] from each node $u$. We record its distance map $d(u, v)$, and define: diameter $= \max_u \max_{v \neq u} d(u, v)$. A large diameter indicates that the reasoning graph explores a wider range of potential reasoning nodes during inference.

**Small-World index**    Small-world organisation is a robust network feature that has been observed in diverse domains—social networks [57, 65], biological and neural systems [28, 53], ecological webs [43], and technological networks such as the World-Wide Web [1]. While the graph diameter represents the maximum geodesic length, it says nothing about local connectivity. We therefore evaluate the *small-world index*. Following Humphries and Gurney [26], we first symmetrise the directed reasoning graph to obtain an undirected neighbour set $\mathcal{N}(i)$ for each node $i$. With $n_i = |\mathcal{N}(i)|$, $N_C = |\{i : n_i \geq 2\}|$, and $N_L = \sum_u \sum_{v \neq u} \mathbf{1}_{\{v \text{ reachable from } u\}}$, we define

$$C_i = \frac{\#\{\text{edges among neighbors of } i\}}{n_i(n_i - 1)/2}, \quad C = \frac{1}{N_C} \sum_{i:n_i \geq 2} C_i, \quad L = \frac{\sum_u \sum_{v \neq u} d(u, v)}{N_L},$$

where $d(u, v)$ is the shortest-path distance from node $u$ to node $v$. Letting $N$ be the total number of nodes and $K$ the mean degree of the undirected graph, we approximate the corresponding

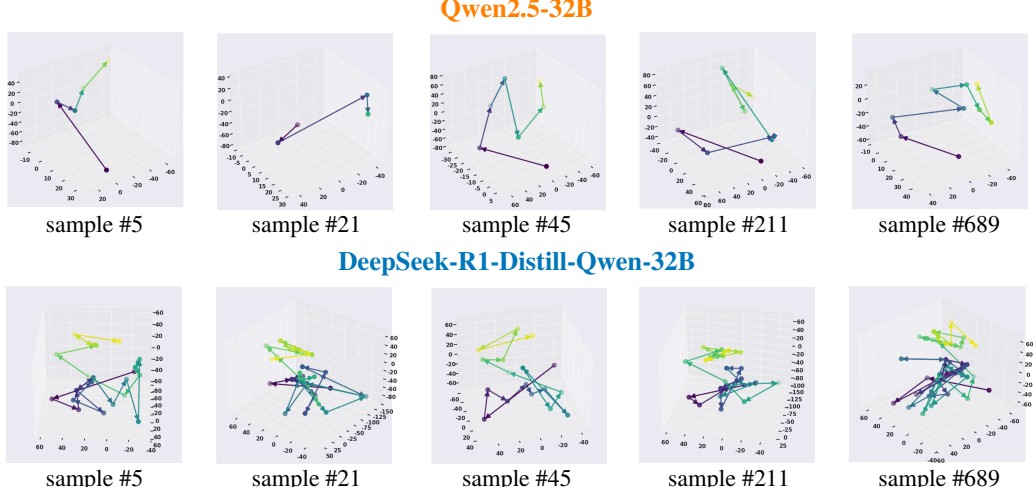

Qwen2.5-32B

| sample #5 | sample #21 | sample #45 | sample #211 | sample #689 |

DeepSeek-R1-Distill-Qwen-32B

| sample #5 | sample #21 | sample #45 | sample #211 | sample #689 |

Figure 3: Visualization of reasoning graphs on GSM8K dataset using t-SNE embeddings. The upper row shows graphs from base model (Qwen2.5-32B), while the lower row represents those from the large reasoning model (DeepSeek-R1-Distill-Qwen-32B). Compared to the base model, the reasoning model exhibits qualitatively broader exploration with notably more cycles in its reasoning graphs.

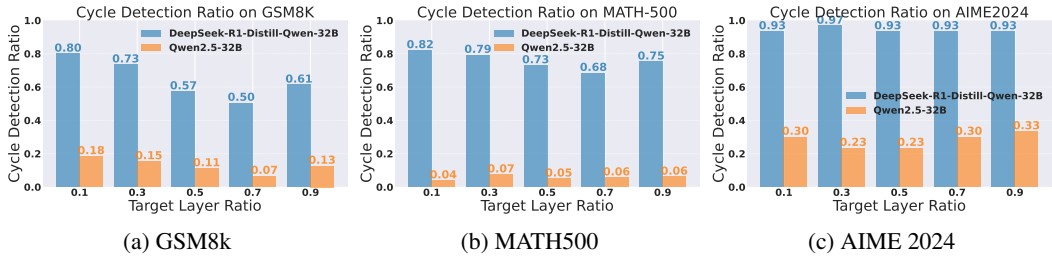

| (a) GSM8k | (b) MATH500 | (c) AIME 2024 |

Figure 4: Comparison of cycle detection ratios across different layers in the large reasoning model (DeepSeek-R1-Distill-Qwen-32B) and the base model (Qwen2.5-32B), evaluated on three tasks: **(a)** GSM8K, **(b)** MATH500, and **(c)** AIME 2024. Results consistently show that the large reasoning model exhibits significantly higher cycle detection ratios than the base model at all layer ratios and tasks. Additionally, a trend emerges, indicating that the cycle detection ratio increases as task difficulty escalates from GSM8K through MATH500 to AIME 2024.

Erdős–Rényi random-graph baseline values [4] as $C_{\text{rand}} = \frac{K}{N-1}$, $L_{\text{rand}} = \frac{\ln N}{\ln K}$, and define the small-world index by $S = \frac{C/C_{\text{rand}}}{L/L_{\text{rand}}}$. The clustering coefficient describes the tendency of nodes to form tightly interconnected groups, while the average path length indicates how efficiently information propagates through the network. The small-world index combines these characteristics, highlighting a graph's ability to maintain local cohesion while supporting rapid global connectivity.

## 4 Analyzing Enhanced Reasoning through Graph-Theoretic Properties

We utilize the Qwen2.5 family distilled from DeepSeek-R1 [10] as our large reasoning models, available in sizes of 1.5B, 7B, 14B, and 32B parameters. The corresponding base models and their details are provided in Appendix D. By default, we use the highest-performing 32B variant. Unless otherwise specified, we extract reasoning graphs from the hidden layer positioned at 90% depth (e.g., layer 58 in the 64-layer 32B model). We employ the GSM8K [7], MATH500 [23], and AIME 2024 [11] datasets for constructing the reasoning graphs. Additional analyses on non-mathematical tasks, including StrategyQA [22] and LogicalDeduction from BIG-Bench [54], are provided in Appendix E.

### 4.1 Visualization and Quantification of Cycles in Reasoning Graphs

To intuitively capture the characteristics of reasoning graphs in large reasoning models, we first visualize reasoning graphs for some samples from the GSM8K dataset using 3-dimensional t-SNE

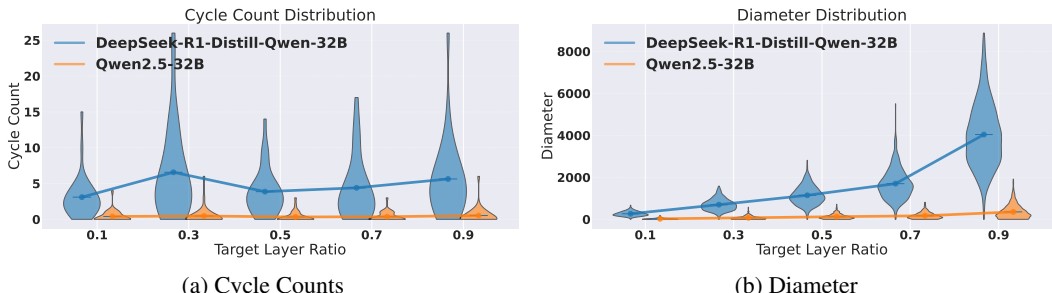

| (a) Cycle Counts | (b) Diameter |

Figure 5: **(a)** Distribution of cycle counts for the large reasoning model (DeepSeek-R1-Distill-Qwen-32B) and the base model (Qwen2.5-32B) across various hidden layer depths. The reasoning model exhibits significantly higher cycle counts. **(b)** Distribution of reasoning graph diameters across various hidden layer depths. The diameter of reasoning graphs increases progressively with deeper layers. The reasoning model demonstrates significantly larger graph diameters, indicating a broader exploration space compared to the base model.

embedding, as depicted in Figure 3. In the visualization, reasoning graphs are represented as directed arrows connecting nodes visited during inference. The base model demonstrates relatively simple and predominantly *acyclic* reasoning graphs. In contrast, the large reasoning model exhibits more complex structures, characterized by frequent *cyclic* patterns and broader node coverage.

To quantitatively validate these qualitative observations, we employed the cycle detection method introduced in Section 3.2. Figure 4 shows cycle detection rates for GSM8K, MATH500, and AIME 2024 datasets, comparing the large reasoning model (blue) with the base model (orange). The horizontal axis denotes different relative depths of hidden layers (0.1, 0.3, 0.5, 0.7, and 0.9), corresponding respectively to layers 6, 19, 32, 45, and 58 in the 64-layer Qwen2.5-32B. Across all layers, the large reasoning model consistently exhibited a notably higher frequency of cyclic reasoning graphs compared to the base model. Additionally, we observed higher cycle detection rates at the earlier and later layers, with lower detection rates in intermediate layers. This pattern suggests that intermediate layers compress token representations, making less cycle detection difficult, whereas layers closer to input or output exhibit clearer cyclic behaviors. The results for varying the hyperparameter $k$ of the $K$-means clustering are provided in Appendix F, showing consistent trends across all tested values of $k$. Furthermore, another consistent trend emerges in which cycle detection ratios increase with the increasing complexity of tasks, progressing from GSM8K through MATH500 to AIME 2024. These findings reinforce the hypothesis that cycles within reasoning graphs contribute to the enhanced reasoning capabilities observed in large reasoning models.

Figure 5-(a) illustrates the distribution of cycle counts per sample. The large reasoning model consistently exhibits higher cycle counts. It indicates that the large reasoning model not only exhibits a higher proportion of samples containing cycles but also features a higher average number of cycles per sample, approximately five cycles on average. These findings also emphasize the importance of cyclic structures in reasoning graphs as a critical characteristic that improves reasoning performance.

## 4.2 Analyzing Reasoning Exploration through Graph Diameter

To better understand exploratory behaviors within reasoning graphs, we analyzed the distribution of graph diameters for both the large reasoning model and the base model using the GSM8K dataset. As shown in Figure 5-(b), the large reasoning model consistently demonstrates larger graph diameters across all examined layers compared to the base model. This indicates that the large reasoning model explores a wider range of reasoning nodes during inference, likely contributing to its superior reasoning performance.

Moreover, we observed a clear trend of progressively increasing graph diameters in deeper hidden layers, suggesting that richer contextual representations at deeper layers correspond to broader exploration scopes. These observations imply similarities between expanding the number of output tokens (thus enlarging the exploration scope) and increasing model depth from the perspective of reasoning graph diameters. Our findings suggest a unified explanation based on reasoning graph diameters, which aligns closely with recent studies that emphasize improved reasoning through iterative deep-layer processing [42, 21]. Additional results for the MATH500 and AIME 2024 datasets can be found in Appendix G, demonstrating the same trends across all tasks.

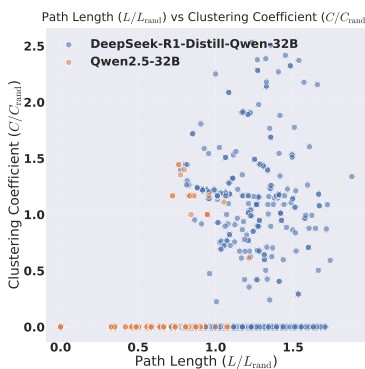

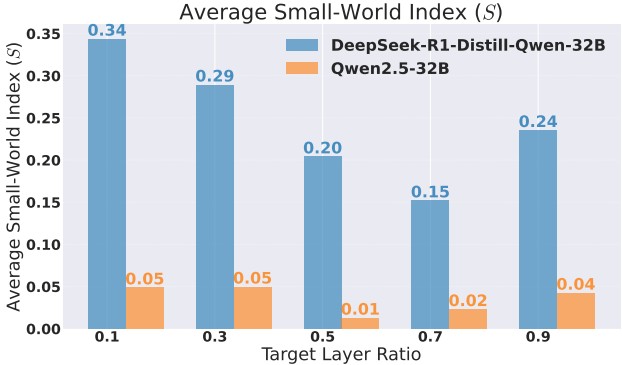

(a) Path Length & Clustering Coefficient    (b) Small-World index

Figure 6: **(a)** Distribution of average path lengths and clustering coefficients in the large reasoning model (DeepSeek-R1-Distill-Qwen-32B) and the base model (Qwen2.5-32B). Reasoning models exhibit larger clustering coefficients and longer path lengths, indicating densely clustered yet widely separated reasoning nodes. **(b)** Comparison of small-world index computed from clustering coefficients and average path lengths. Across all layers, the reasoning model consistently exhibits higher small-world characteristics compared to the base model.

## 4.3 Small-World Structure and Enhanced Reasoning Efficiency

To gain deeper insights into the graph characteristics underlying reasoning graphs, we examine their small-world properties. Specifically, using the AIME 2024 dataset, we examine clustering coefficients ($C$), average path lengths ($L$), and their relationship through the Small-World Index ($S$).

Figure 6-(a) depicts the distributions of clustering coefficients and average path lengths for the large reasoning model and the base model at a layer ratio of 0.9. The large reasoning model clearly demonstrates notably higher clustering coefficients alongside longer average path lengths. This combination indicates that reasoning graphs in large reasoning models form densely interconnected local clusters while also having some nodes connected by relatively long-range paths. Such a structure allows quick access via short paths to arbitrary nodes within local clusters, facilitating easier recovery from incorrect reasoning pathways and potentially enhancing reasoning performance. This structural pattern aligns with recent theoretical findings [31] that model reasoning processes as Markov chains, comprising densely connected nodes (representing simple reasoning steps) and sparsely connected critical transitions (representing complex reasoning steps). Additional results for other layer ratios are provided in Appendix H, consistently showing similar trends across all layers.

In Figure 6-(b), we further present the Small-World Index ($S$) computed across different hidden layer depths. The large reasoning model consistently shows higher $S$ values compared to the base model across all layers. Interestingly, we observe a declining trend in $S$ near intermediate layers, reflecting the previously discussed cyclic reasoning behaviors. Collectively, these findings underscore the essential contribution of small-world graph characteristics to advanced reasoning performance and suggest valuable avenues for future theoretical and empirical exploration.

## 4.4 Impact of Model Size on Reasoning Graph Properties

To clarify how model size influences reasoning graph properties, we analyzed the relationship between cycle detection ratios, reasoning graph diameters, cycle counts, and task accuracy on the AIME 2024 dataset. The results of the small-world index relative to model size are detailed in Appendix I.

Figure 7-(a) demonstrates that the cycle detection ratio generally increases with model size, peaking at a 100% cycle detection rate in the 14B model. Interestingly, our largest 32B model, which achieves the highest task accuracy, exhibits a lower cycle detection ratio than the 14B model. To better understand the reason for this trend, we compares the outputs generated by the 14B and 32B models in Figure 8. We find that the 14B model experiences *language mixing*, a phenomenon recently reported [68] where language model switches and repeat different languages during the reasoning process. Such undesirable cyclicity likely explains why the 14B model, despite its higher cycle detection ratio, underperforms relative to the 32B model.

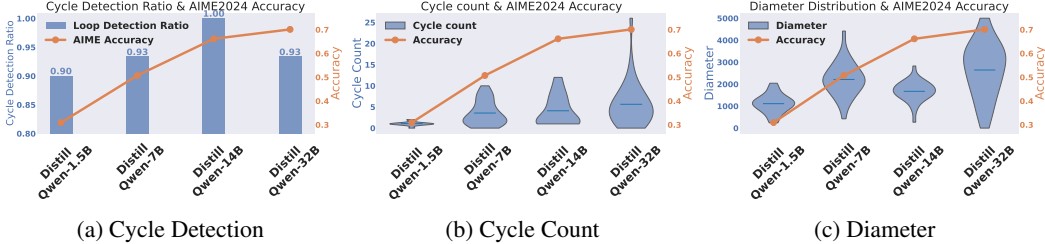

|(a) Cycle Detection|(b) Cycle Count|(c) Diameter|

Figure 7: **(a)** Relationship between cycle detection ratio and AIME 2024 accuracy across different model sizes. The cycle detection ratio generally increases with model size up to 14B, which achieves a 100% cycle detection ratio. **(b)** Relationship between cycle count and accuracy across different model sizes. Larger models demonstrate increased cycle counts, with the 32B model, which achieves the highest accuracy, exhibiting the greatest number of cycles. **(c)** Relationship between reasoning graph diameter and accuracy across different model sizes. The 32B model, achieving the highest accuracy, also exhibits the largest graph diameter.

These findings indicate that although cyclic reasoning generally enhances reasoning effectiveness, certain types of cycles, such as language mixing, do not positively contribute to performance.

Figure 7-(b) illustrates that cycle counts progressively rise with model size, reaching the maximum in the 32B model. The observed positive correlation suggests that iterative revisitation of reasoning nodes fosters deeper refinement, thereby substantially enhancing performance in complex math tasks. Figure 7-(c) highlights that the reasoning graph diameter expands with model size, with the largest (32B) model consistently showing the most significant diameters alongside the highest accuracy. This suggests that broader exploration and complex reasoning paths are crucial for superior reasoning performance.

Collectively, these results imply that larger model capacities enable greater cyclicity and wider exploration within

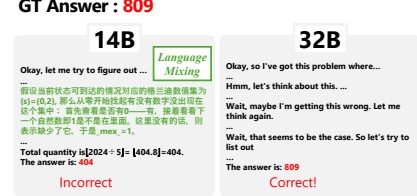

Figure 8: Comparison of reasoning outputs from the 14B and 32B models. The 14B model exhibits *language mixing* [68], switching languages through the response, while the 32B model maintains a consistent language.

reasoning graphs, aligning with prior analyses showing smaller models struggle more in reasoning tasks [36].

## 5 Evolution of Reasoning Graph Properties during Supervised Fine-Tuning

To bridge the graph-theoretic properties of large reasoning models observed in Section 4 with practical improvements in reasoning performance, we conducted SFT using the s1 dataset [44], which significantly enhances the reasoning capabilities of the Qwen2.5-32B-Instruct. We analyzed the evolution of reasoning graph properties across training steps to elucidate how these characteristics emerge through training. Detailed training parameters are provided in Appendix J. Our experiments utilized two versions of the dataset: the original version (s1-v1.0 [1]) and an updated version (s1-v1.1 [2]), each consisting of 1000 training samples. Performance metrics on benchmark datasets indicated higher efficacy for the updated dataset, with v1.1 achieving 94.4% accuracy compared to 92.6% for v1.0 on MATH500, and 56.7% versus 50.0% accuracy on AIME 2024, respectively [44].

Figure 9-(a) illustrates the differences in reasoning graph diameters between s1-v1.0 and s1-v1.1 at 200 training steps, and Figure 9-(b) at 400 training steps across various layers on AIME 2024. It can be observed that, on average, s1-v1.1 consistently produces larger diameters across all layers. Furthermore, there are more samples exhibiting larger diameters at 400 steps compared to 200 steps. These findings suggest that the diameter of reasoning graphs is amplified by SFT, and notably, superior SFT data such as s1-v1.1 enhances the reasoning graph diameter, effectively expanding the exploration space. The results for other checkpoints are provided in Appendix K.

---

[1]https://huggingface.co/datasets/simplescaling/s1K
[2]https://huggingface.co/datasets/simplescaling/s1K-1.1

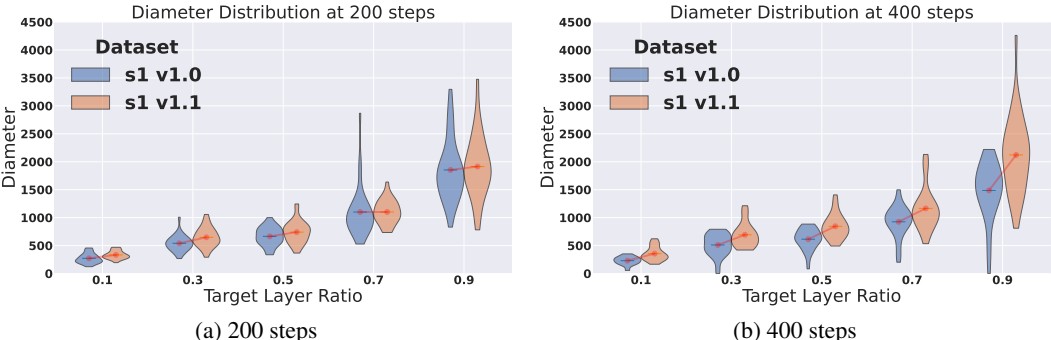

Figure 9: Comparison of reasoning graph diameter distributions between datasets s1-v1.0 and s1-v1.1 across different hidden layers at **(a)** 200 training steps and **(b)** 400 training steps. Dataset s1-v1.1 consistently yields larger graph diameters compared to s1-v1.0, and graph diameters increase as training progresses from 200 to 400 steps, indicating enhanced exploration capacity facilitated by superior SFT data.

# 6 Discussion

In this work, we conducted an extensive analysis of reasoning graphs derived from large reasoning models, uncovering key structural properties that correlate with their enhanced performance. Our main findings highlight that large reasoning models consistently exhibit (1) greater cyclicity, (2) broader exploratory behaviors (larger diameters), and (3) pronounced small-world characteristics compared to base models. These insights suggest sophisticated structures in reasoning graphs as a critical factor driving reasoning improvements. Our results connect several observed behaviors in large reasoning models and offer implications for constructing more effective training datasets.

**Aha Moment**  Models trained via RL have been reported to exhibit an intriguing phenomenon known as the *"aha moment,"* where the model reconsidered its intermediate answers during reasoning [10, 68]. From the perspective of our reasoning graph analysis, this phenomenon aligns consistently with the observed *cyclic* structures (as illustrated in Figure 1). Although the *"aha moment"* was initially identified as a phenomenon at the generated token level, our study quantitatively measures this behavior through the *cycle* properties of reasoning graphs, thereby contributing to a deeper mechanistic understanding of the *"aha moment"* from the internal states of LLMs.

**Overthinking and Underthinking**  Recent studies have highlighted specific reasoning inefficiencies in large reasoning models. *Overthinking*, characterized by redundant or excessively long reasoning processes, has been frequently observed, particularly in agent-based tasks [32, 55, 8, 14]. Conversely, models in the o1 family display *underthinking*, rapidly switching thoughts without adequately exploring potentially valuable reasoning paths [64]. These phenomena align closely with the graph properties we have analyzed: redundant cyclic structures (discussed in Section 4.4) explain overthinking, while overly extensive exploratory behaviors (reflected in larger graph diameters, discussed in Section 4.2) may account for underthinking. Thus, our research clarifies these unique behaviors of large reasoning models through the lens of reasoning graph characteristics.

**Implications for Reasoning SFT-Data Construction**  Some studies have significantly improved reasoning performance through SFT or DPO [49, 15] with limited data [72, 44, 69]. While these studies typically create datasets based on qualitative criteria such as difficulty, quality, and diversity, which are inherently challenging to quantify, our proposed metric based on reasoning graph characteristics extracted from hidden states provide novel insights for dataset construction. For example, the high-quality dataset s1-v1.1 demonstrated notably larger graph diameters, suggesting its structural properties are indicative of superior reasoning potential. Furthermore, as shown in Appendix L, even when comparing the s1-v1.0 against LIMO, the s1 dataset consistently yields reasoning graphs with larger diameters and more cycles across all layer depths. This suggests that higher-quality SFT data induces more exploratory and reflective reasoning behavior, further supporting the use of graph-theoretic metrics as indicators of data effectiveness.

## Acknowledgements

We thank Heiga Zen and Yinlam Chow for their support and reviewing the draft of this paper. We also appreciate the funding support from Google Japan.

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

# A  Broader Impacts

This research sheds light on the underlying mechanisms responsible for improved reasoning performance in LLMs, potentially impacting various fields within artificial intelligence and machine learning. From an interpretability perspective, our findings offer explanations for performance gains in previously opaque (black-box) large reasoning models and provide insights toward the construction of better reasoning architectures.

# B  Automatic Node Labeling with LLMs

To better understand what kinds of reasoning patterns are represented by the centroids when $k = 200$, we conducted an automatic labeling experiment using a large language model. Specifically, we used the `GPT-4o-mini` [46] API to assign a theme to each centroid, based on the reasoning steps associated with it in the reasoning graph of DeepSeek-R1-Distill-Qwen-32B on the GSM8K dataset.

We provided the following system prompt to GPT, and then input multiple reasoning steps corresponding to each centroid as the user prompt:

```
You are a data analyst.  The following is an output from a
LLM.
Your task is to carefully read the text and summarize its main
theme in 1-3 English words.
```

Table 1 presents the assigned **theme** for each centroid, together with **example** reasoning steps that were mapped to that centroid. This analysis revealed that many centroids align with interpretable reasoning patterns. In addition to previously reported in Figure 2 such as *Add*, *Multiply*, and *Wait*, we identified centroids associated with higher-level computations (e.g., *Calculations Totals*, *Averages*), semantics-bound reasoning (e.g., *Age Calculations*, *Cost Calculations*), and structural elements (e.g., *Answer Formatting*, *Placeholder Tags*).

We also observed centroids linked to reasoning behaviors, such as *Planning*, which reflects the model's initial steps when approaching a math problem. Moreover, instances of *Wait* appear to split into multiple subtypes. For example, centroids like *Calculation Correctness* and *Reevaluation* capture the model's tendency to reassess or double-check its output. A distinct centroid labeled *Inconsistencies* highlights cases where the model detects contradictions in its reasoning and attempts to revise its calculations.

Overall, these results indicate that the centroids discovered by the reasoning graph clustering procedure correspond to a wide variety of meaningful and interpretable reasoning techniques.

Table 1: Examples of automatically identified themes and corresponding reasoning steps.

| Theme (Node id) | Examples of Reasoning Steps |
| --- | --- |
| **Calculations — Totals (node 83)** | • 5 + 10 = 15. Then, 15 + 9 = 24. Finally, 24 + 3 = 27.
• Let me add them step by step. 500 + 1500 = 2000. Then, 2000 + 125 = 2125. |
| **Calculations — Average (node 119)** | • Average = (Sum of all values) / (Number of values).
• Average Speed = Total Distance / Total Time = 250 miles / 5 hours = 50 mph. |
| **Calculations — Division (node 41)** | • 125,000 / 20 = 6,250.
• 120 pieces / (15 pieces per pack) = 8 packs.
• 80 / 10 = 8 weeks. |
| **Age Calculations (node 147)** | • Sum of their ages in two years = (B + 2) + (2B + 2) = 28.
• C = 2 × (James's age in 8 years) − 5. |
| **Cost Calculations (node 168)** | • $47.00 × 5 = $235.00.
• Keenan's weekly cost = $160 ÷ 4 = $40.
• 8 × 8 = $64.00. |
| **Answer Format (node 137)** | • The answer is: 50.
• The answer is: 8. |
| **Placeholder Tags (node 26)** | • `</think>` |
| **Planning (node 163)** | • Okay, so I need to figure out how much Leah has spent on her new kitten so far. Let me break it down step by step.
• Hmm, let's break it down step by step. |
| **Calculation Correctness (node 52)** | • Wait, that seems straightforward, but let me double-check. . .
• Wait, let me double-check my calculations to make sure. . . |
| **Reevaluation (node 100)** | • Wait, maybe I made a mistake in the equations.
• Wait, maybe I made a mistake in the equations. Let me try to model it again. . . |
| **Inconsistencies (node 110)** | • Wait, that's a problem. 15 + 8 = 23, which is more than total time 20.
• Wait, perhaps the shows are part of the 30%, and the other activities are part of the remaining 70%. But that doesn't make sense. . . |

# C  Measuring the Graph Property Implementation

Here is a sketch of Python code for detecting cycles and computing the diameter of reasoning graphs:

```python
from collections import defaultdict, deque
import heapq

def analyze_graph_simple(path, distances):
    adj = defaultdict(list)
    for u, v, w in zip(path, path[1:], distances):
        if u != v:
            adj[u].append((v, w))
    # Cycle detection
    seen, has_loop = set(), False
    loop_count = 0
    entry_node = None
    for i, node in enumerate(path):
        if node in seen:
            has_loop = True
            entry_node = node
            loop_count = path.count(node) - 1
            break
        seen.add(node)

    # Diameter and Avg Path Length
    def dijkstra(u):
        dist = {u: 0}
        heap = [(0, u)]
        while heap:
            d, node = heapq.heappop(heap)
            for neighbor, weight in adj[node]:
                new_dist = d + weight
                if neighbor not in dist or new_dist < dist[neighbor]:
                    dist[neighbor] = new_dist
                    heapq.heappush(heap, (new_dist, neighbor))
        return dist

    all_distances = [dijkstra(node) for node in adj]
    diameter = max((max(dists.values()) for dists in all_distances), default=0)
    avg_path_length = \
    sum(sum(dists.values()) for dists in all_distances) / sum(len(dists)-1 for dists in all_distances)

    # Clustering Coefficient
    undirected = defaultdict(set)
    for u, neighbors in adj.items():
        for v, _ in neighbors:
            undirected[u].add(v)
            undirected[v].add(u)

    clustering_sum, count_cc = 0, 0
    for node, nbrs in undirected.items():
        if len(nbrs) < 2:
            continue
        actual_edges = sum(1 for v in nbrs for w in nbrs if v < w and w in undirected[v])
        clustering_sum += actual_edges / (len(nbrs) * (len(nbrs)-1) / 2)
        count_cc += 1
    avg_clustering = clustering_sum / count_cc if count_cc else 0

    # Small-World Index
    N = len(undirected)
    K = sum(len(nbrs) for nbrs in undirected.values()) / N if N else 0
    C_rand = K / (N - 1) if N > 1 else 0
    L_rand = math.log(N) / math.log(K) if N > 1 and K > 1 else float('inf')

    clustering_norm = avg_clustering / C_rand if C_rand else 0
    path_length_norm = avg_path_length / L_rand if L_rand else 0
    small_world_index = clustering_norm / path_length_norm if path_length_norm else 0

    return has_loop, loop_count, diameter, avg_clustering, avg_path_length, small_world_index
```

# D  Details of Large Reasoning Model and Base Model

For the large reasoning models, we utilize distilled models from the DeepSeek-R1 series, which are derived from Qwen-based models. Correspondingly, their base models are the original Qwen and Llama models prior to distillation. A detailed list of these models is provided in Table 2.

Table 2: Comparison of large reasoning models and corresponding base models.

| Base Model | Large Reasoning Model |
| --- | --- |
| Qwen2.5-Math-1.5B[1] | DeepSeek-R1-Distill-Qwen-1.5B[2] |
| Qwen2.5-Math-7B[3] | DeepSeek-R1-Distill-Qwen-7B[4] |
| Llama-3.1-8B[5] | DeepSeek-R1-Distill-Llama-8B[6] |
| Qwen2.5-14B[7] | DeepSeek-R1-Distill-Qwen-14B[8] |
| Qwen2.5-32B[9] | DeepSeek-R1-Distill-Qwen-32B[10] |

# E  Extension to Non-Math Reasoning Tasks

To examine whether the observed properties of reasoning graphs generalize beyond mathematical reasoning, we extended our evaluation to include two additional tasks inspired by prior work [63]: (i) **StrategyQA** [22], a multi-hop question answering dataset, and (ii) **LogicalDeduction**, a logical reasoning dataset from BIG-Bench [54]. In Figure 10, we compare reasoning graph properties between DeepSeek-R1-Distill-Qwen-32B (reasoning model) and Qwen2.5-32B (base model) on these datasets. Similar to the math tasks reported in the main paper, the reasoning model exhibits markedly different graph characteristics, including higher cycle rates, larger diameters, and stronger small-world properties. These consistent patterns across multiple domains provide additional evidence that the reasoning graph framework captures structural properties in a general manner.

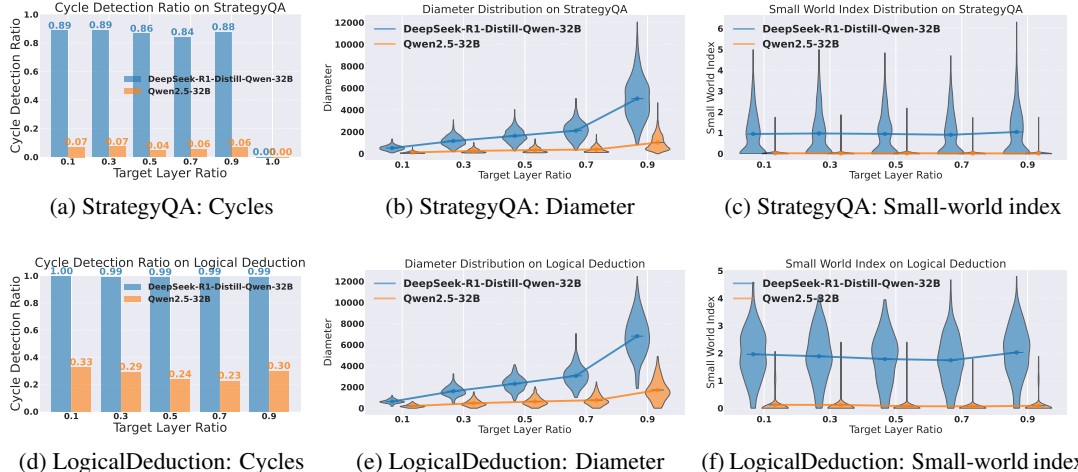

(a) StrategyQA: Cycles    (b) StrategyQA: Diameter    (c) StrategyQA: Small-world index

(d) LogicalDeduction: Cycles    (e) LogicalDeduction: Diameter    (f) LogicalDeduction: Small-world index

Figure 10: Comparison of reasoning graph properties between Distill-Qwen-32B and Qwen2.5-32B on two non-math reasoning datasets: (top row) StrategyQA and (bottom row) LogicalDeduction. Columns correspond to Cycles, Diameter, and Small-world index.

---

[1] https://huggingface.co/Qwen/Qwen2.5-Math-1.5B

[2] https://huggingface.co/deepseek-ai/DeepSeek-R1-Distill-Qwen-1.5B

[3] https://huggingface.co/Qwen/Qwen2.5-Math-7B

[4] https://huggingface.co/deepseek-ai/DeepSeek-R1-Distill-Qwen-7B

[5] https://huggingface.co/meta-llama/Llama-3.1-8B

[6] https://huggingface.co/deepseek-ai/DeepSeek-R1-Distill-Llama-8B

[7] https://huggingface.co/Qwen/Qwen2.5-14B

[8] https://huggingface.co/deepseek-ai/DeepSeek-R1-Distill-Qwen-14B

[9] https://huggingface.co/Qwen/Qwen2.5-32B

[10] https://huggingface.co/deepseek-ai/DeepSeek-R1-Distill-Qwen-32B

# F  Experiments with Different $K$ Values in $K$-Means Clustering

We report differences in cycle detection ratios when varying the number of clusters in $K$-means clustering, specifically for the GSM8K dataset, as shown in Figure 11. As expected, decreasing results in fewer clusters and a higher ratio of detected cycles. Across all values, the large reasoning model consistently exhibits a higher cycle ratio compared to the base model.

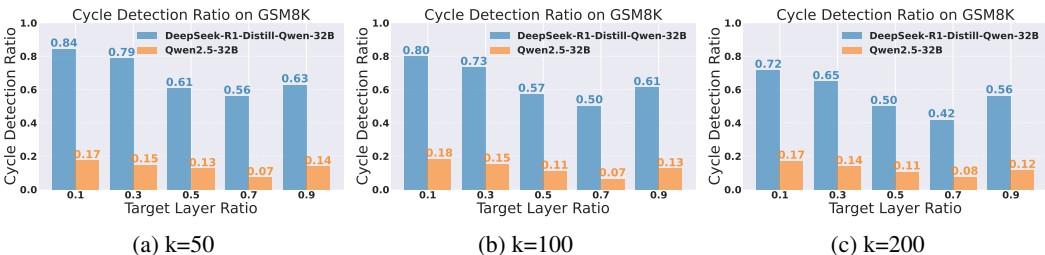

Figure 11: Comparison of cycle detection ratios across different layers for the large reasoning model (DeepSeek-R1-Distill-Qwen-32B) and the base model (Qwen2.5-32B), evaluated on three tasks: **(a)** $k = 50$, **(b)** $k = 100$, and **(c)** $k = 200$. Results consistently show that the large reasoning model exhibits significantly higher cycle detection ratios than the base model at all layer ratios and $k$.

# G  Diameter Analysis on MATH500 and AIME 2024

Figure 12 compares reasoning graph diameters for the MATH500 and AIME 2024 datasets. The large reasoning model consistently exhibits greater diameters than the base model, with a clear trend of increasing diameter in deeper hidden layers, aligning with observations from the GSM8K dataset.

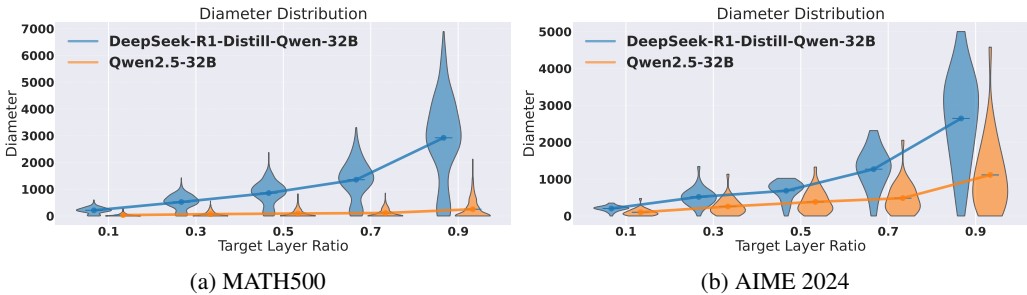

Figure 12: Diameter of Reasoning Graph in DeepSeek-R1-Distill-Qwen-32B and Qwen2.5-32B.

## H  Layer-wise Clustering Coefficient and Average Path Length

Figure 13 shows the clustering coefficient and average path length for each layer. The large reasoning model consistently exhibits higher clustering coefficients at all layer ratios, contributing to its enhanced small-world characteristics.

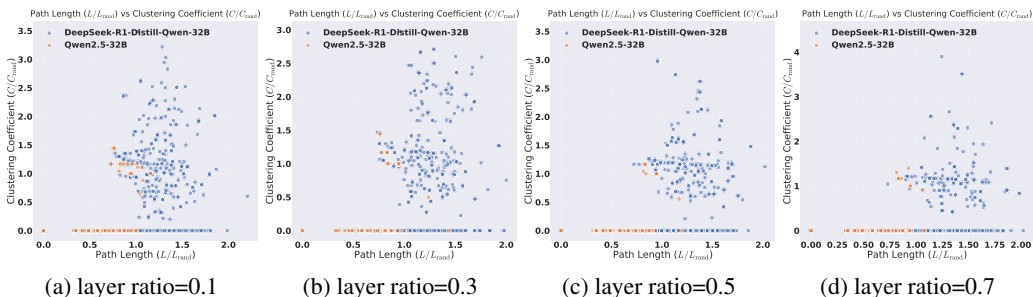

(a) layer ratio=0.1  (b) layer ratio=0.3  (c) layer ratio=0.5  (d) layer ratio=0.7

Figure 13: The clustering coefficient and average path length for each layer.

## I  Model Size and Small-World Index

Figure 14 presents the relationship between model size and the Small-World Index. The results suggest that the Small-World property becomes more pronounced as the model size increases.

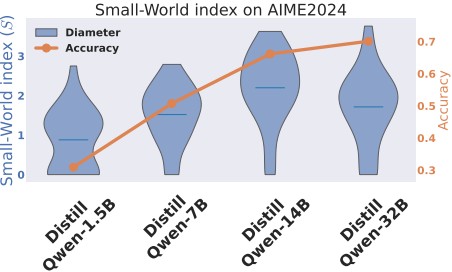

Figure 14: Relationship between model size and the Small-World Index.

## J Training Details

We conducted supervised fine-tuning (SFT) experiments using the Qwen2.5-32B-Instruct model as our base model. The training was executed on a computing node equipped with 8 NVIDIA H200 GPUs for training and a single NVIDIA H200 GPU for inference.

The detailed training configuration is summarized in Table 3.

Table 3: Detailed training configuration for the SFT experiments.

| Parameter | Value |
|---|---|
| Base Model | Qwen2.5-32B-Instruct |
| Dataset | simplescaling/s1K or simplescaling/s1K-1.1 |
| Number of Epochs | 5 |
| Learning Rate | $1 \times 10^{-5}$ |
| Learning Rate Scheduler | Cosine (minimum LR: 0) |
| Batch Size | 8 (Effective: 8 GPUs $\times$ micro-batch size 1) |
| Gradient Accumulation Steps | 1 |
| Weight Decay | $1 \times 10^{-4}$ |
| Optimizer | AdamW ($\beta_1 = 0.9$, $\beta_2 = 0.95$) |
| Warmup Ratio | 0.05 |
| Precision | bf16 |
| Gradient Checkpointing | Enabled |
| FSDP Configuration | Full Shard Data Parallel (auto-wrap) |
| Block Size | 32768 tokens |

Inference was conducted using a single NVIDIA H200 GPU to evaluate trained models and generate results presented in the paper.

## K Diameter of other s1 checkpoints

Figure 15 compares the diameters of reasoning graphs at 100 and 500 training steps using the S1 dataset [44]. In both cases, version s1-v1.1 demonstrates larger diameters compared to s1-v1.0, and the diameters tend to increase with further training steps.

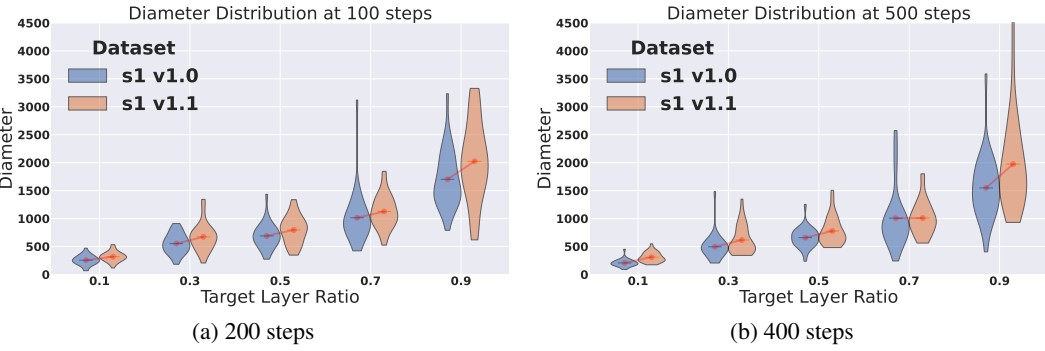

(a) 200 steps                                    (b) 400 steps

Figure 15: Cycle Graph Detection Ratio in DeepSeek-R1-Distill-Qwen-32B and Qwen2.5-32B.

## L  Impact of SFT Data Quality on Reasoning Graph Structure.

To further investigate the relationship between data quality and reasoning-graph properties, we compared two supervised fine-tuning (SFT) datasets: LIMO [72] and s1 v1.0 [44]. The s1 dataset has previously been recognized for its strong performance in enhancing reasoning abilities [44].

We constructed reasoning graphs from the hidden states of Qwen2.5-32B-Instruct, prompted with data from s1 and LIMO (note that we did not fine-tune Qwen2.5-32B-Instruct on these datasets).

As shown in Figure 16, reasoning graphs derived from s1 consistently exhibit larger diameters and higher cycle counts across all examined layer depths. This indicates that the s1 dataset inherently induces exploration of a broader range of latent reasoning states, resulting in more iterative reasoning. In contrast, graphs derived from LIMO show narrower reasoning trajectories with fewer cycles, suggesting more linear and potentially shallow reasoning processes.

These findings suggest that higher-quality SFT data possess richer reasoning-graph characteristics (such as increased cycles and larger diameters), which in turn contribute to improved performance when used for fine-tuning. Thus, reasoning-graph analysis from hidden states offers a novel perspective and practical guidance for creating better SFT datasets.

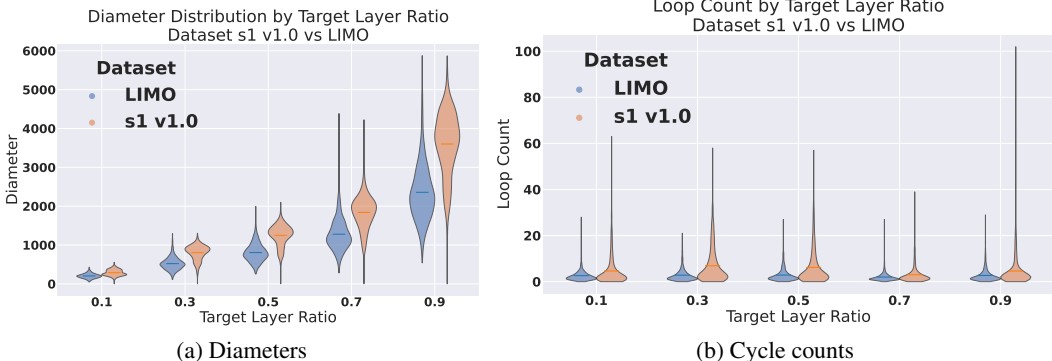

(a) Diameters     (b) Cycle counts

Figure 16: We constructed reasoning graphs from the hidden states of Qwen2.5-32B-Instruct, prompted with data from s1 and LIMO (note that we did not fine-tune Qwen2.5-32B-Instruct on these datasets). The s1 dataset—regarded as higher-quality—consistently yields larger diameters and higher cycle counts, indicating broader exploration and more reflective reasoning behavior.

## M  Limitations

This study introduces the concept of reasoning graphs as a tool to identify distinctive graph-theoretic properties that may explain recent breakthroughs in the reasoning performance of large language models (LLMs). While our findings provide a novel explanatory perspective on the reasoning capabilities of advanced models, concrete guidelines on constructing models with superior reasoning performance remain insufficient. Although we experimentally examine the relationship between graph properties and reasoning-SFT in Section 5, using these insights as a first step toward building more effective reasoning models is left for future work.

Our analysis focuses on transitions in the context direction of hidden states, but it does not provide feature-level [25, 56, 20, 37, 39] or circuit-level analyses [45, 61, 40] as commonly studied in mechanistic interpretability [3, 52]. Furthermore, how the distinctive graph properties observed in reasoning models emerge during training dynamics [16, 60, 41] remains an open question, which we leave as an important direction for future work.

