# OpenReview forum: "Topology of Reasoning: Understanding Large Reasoning Models through Reasoning Graph Properties"
_NeurIPS.cc/2025/Conference — NeurIPS 2025 poster_

### Official Review · Reviewer_EMKy · 2025-06-23

**Clarity:** 3
**Significance:** 2
**Originality:** 3
**Rating:** 3
**Confidence:** 4

**Summary:**

This paper introduces reasoning graphs, which are constructed by clustering hidden states of LLMs during reasoning tasks. It analyzes their graph-theoretic properties (cyclicity, diameter, small-world index), demonstrating that reasoning models exhibit significantly more cycles, larger exploration diameters, and stronger small-world characteristics compared to base models. The work provides new insights into the structural mechanisms behind advanced reasoning capabilities and practical guidance for data design.

**Questions:**

1. The authors should conduct additional experiments beyond mathematical reasoning datasets to demonstrate the broader applicability of their findings. (W1)
2. The authors must resolve the issues stated in W2-W4.
3. The authors should explain why they chose the hidden states from the layer of 90% depth as the default rather than the last layer. The hidden states of the final layer are used to generate responses, which seems significantly more reasonable.
4. The authors could demonstrate the statistics (e.g., average number of nodes and edges) of the reasoning graphs for each dataset to help readers better understand the changes in graph properties.

**Ethical Concerns:**

["NO or VERY MINOR ethics concerns only"]

**Final Justification:**

The authors have addressed some of my concerns; however, the issue regarding the practical usability of the proposed method has not been sufficiently resolved. Therefore, I will maintain my score.

**Limitations:**

yes

**Paper Formatting Concerns:**

The table titles, such as Table 1 and Table 2, are not followed by one line space as required in the instructions.

**Quality:**

2

**Strengths And Weaknesses:**

**Strengths:**
1. This paper is well-organized and easy to follow.
2. The idea of extracting reasoning graphs from LLM and exploring their properties is interesting, offering a new perspective for understanding the model's reasoning mechanisms

**Weaknesses:**
1. The conclusions of this paper are only validated by three math reasoning datasets, leaving their generalization to a broader domain (such as logical reasoning, code reasoning, and general tasks) uncertain.
2. The paper emphasizes that the proposed method can guide the design of training data to enhance reasoning capabilities, but it fails to explicitly specify how to construct such data in practice. Moreover, this approach resembles a post-hoc analysis that incurs prohibitive computational costs—since it requires inference across multiple datasets before yielding actionable insights—severely limiting its real-world applicability.
3.  I am not convinced that the approach proposed in the paper—selecting better datasets solely based on the diameter of the reasoning graph—is practically effective. Taking the same scenario as presented in the [S1 paper](https://arxiv.org/abs/2501.19393) as an example: when a large amount of supervised data has been collected, how can the authors' method be applied to select samples for constructing an optimal dataset?
4. The logic underlying the paper’s proposed method for constructing SFT data is flawed. The paper only empirically shows that SFT data with more cycles and larger diameters in the reasoning graph tend to yield better performance. However, the reverse does not necessarily hold: SFT data with more cycles and larger diameters do not inherently lead to better performance. Intuitively, data that reflects overthinking or unnecessarily long reasoning paths may result in graphs with more cycles and larger diameters, but such data does not guarantee improved outcomes. As noted in the S1 paper, selecting data with the longest reasoning traces did not lead to optimal performance.
5. The attribution of redundant cycles to overthinking (Lines 296-298) conflicts with the empirical analysis in Section 4.4, which conclusively traces their origin to language-mixing effects.
6. The paper has many minor flaws that may affect the reader's impression:
    * There is a mismatch between the question and the corresponding response in Figure 1, that is, the generated answers do not address the displayed question.
    * There is a mixed use of notations, such as $K$ in lines 118, 152, and 182, which may lead to reader confusion/misinterpretation.
    * The quality of Figure 3 may hinder proper interpretation, as it is not provided in vector format, resulting in potential pixelation or blurring when scaled.
    * There is an inconsistency between the textual description (Line 641) and Figure 14's caption: The text states the model was fine-tuned using "100 and 500 steps". However, Figure 14's sublabels show results for "200 steps" and "400 steps".
    * There appears to be an issue with invalid citations in Lines 856 and 886, where the references are displayed as question marks ("?").

---

> ### Author Rebuttal · Authors · 2025-07-31
>
> We sincerely thank the reviewer for their thorough and constructive feedback, which significantly helped us strengthen both the empirical scope and clarity of our paper.
>
> **>W1&Q1**
> > The conclusions of this paper are only validated by three math reasoning datasets
>
> To address your concern, we have extended our evaluation to include two additional tasks beyond math, inspired by prior work such as [1]: (i) **StrategyQA**, a multi-hop question answering dataset, and (ii) **LogicalDeduction**, a logical reasoning dataset from BIG-Bench.
>
> Below, we report the comparison of reasoning graph properties between DeepSeek-R1-Distill-Qwen-32B (reasoning model) and Qwen2.5-32B (base model) on these datasets.
> As with the results reported on math tasks in the main paper, the reasoning model exhibits markedly different graph characteristics, such as significantly higher cycle rates, larger diameters, and stronger small-world properties.
>
> These consistent patterns across several domains provide further evidence for the generality of our findings. We appreciate the reviewer’s suggestion, which helped us strengthen the empirical validation of our claims.
>
> ---
> **(i) StrategyQA**
>
> ---
> **Cycles**
> | Model / Layer Ratio | 0.1| 0.3| 0.5| 0.7| 0.9|
> |---|---|---|---|---|---|
> |Distill-Qwen-32B| 0.89| 0.89| 0.86| 0.84| 0.88|
> |Qwen2.5-32B|0.07|0.07|0.04|0.06|0.06|
>
> ---
> **Diameter** (mean±var)
> |Model / Layer Ratio|0.1|0.3|0.5|0.7|0.9|
> |---|---|---|---|---|---|
> |Distill-Qwen-32B|517.60 ± 14.56|1161.89 ± 21.35|1627.66 ± 24.71|2110.36 ± 28.09|5037.20 ± 43.71|
> |Qwen2.5-32B|114.58 ± 8.91|256.53 ± 13.12|335.62 ± 15.06|385.42 ± 16.26|1023.93 ± 25.95|
>
> ---
> **Small world index**
> |Model / Layer Ratio|0.1|0.3|0.5|0.7|0.9|
> |---|---|---|---|---|---|
> |Distill-Qwen-32B|0.94|0.97|0.95|0.91|0.94 |
> |Qwen2.5-32B|0.02|0.02|0.02| 0.01|0.01||
>
> ---
> **(ii) LogicalDeduction**
>
> ---
> **Cycle**
> |Model / Layer Ratio|0.1|0.3|0.5|0.7|0.9|
> |---|---|---|---|---|---|
> |Distill-Qwen-32B|1.00|0.99|0.97|0.99|0.99|
> |Qwen2.5-32B|0.07|0.07| 0.04|0.06|0.06|
>
> ---
> **Diameter** (mean±var)
> | Model / Layer Ratio|0.1|0.3|0.5|0.7|0.9|
> |---|---|---|---|---|---|
> |Distill-Qwen-32B|630.10 ± 13.07| 1608.34 ± 21.33 | 2313.50 ± 25.47 | 3063.10 ± 30.35 | 6815.19 ± 43.48 |
> |Qwen2.5-32B | 210.75 ± 11.69 | 469.29 ± 17.34| 629.74 ± 20.05| 755.24 ± 22.08| 1719.19 ± 32.05|
>
> ---
> **Small world index**
> | Model / Layer Ratio|0.1|0.3| 0.5|0.7|0.9|
> |---|---|---|---|---|---|
> | Distill-Qwen-32B|0.96|0.88| 0.78| 0.74| 0.92|
> | Qwen2.5-32B |0.12|0.11|0.07|0.06|0.08|
>
> [1] Understanding the Reasoning Ability of Language Models From the Perspective of Reasoning Paths Aggregation, ICML'24
>
> **>W2&W3&Q2**
> > it fails to explicitly specify how to construct such data in practice.
>
> > when a large amount of supervised data has been collected, how can the authors' method be applied to select samples for constructing an optimal dataset?
>
> To directly address this concern, and following Reviewer 8fWp’s suggestion, we extended our approach to analyze the reasoning graph during the forward pass of a frozen LLM (Qwen2.5-32B-Instruct, the same model used in S1). Specifically, **we fed various SFT datasets into the model and measured the resulting reasoning graph diameters** without requiring any training.
>
> In this experiment, we used two versions of the S1 dataset: S1-v1.0 and S1-v1.1, where it is known that models trained on S1-v1.1 achieve better downstream reasoning performance. The table below compares the diameters of reasoning graphs induced by these datasets.
> We find that S1-v1.1 consistently yields larger graph diameters, even before any training has occurred, suggesting that the diameter of the reasoning graph can serve as a useful signal for evaluation of SFT data quality before any model training.
>
> This provides a concrete answer to your question: in scenarios like S1, where a large amount of supervised data is available, our method allows one to feed candidate subsets into the target model, construct the corresponding reasoning graphs, and use their properties (e.g., diameter) to guide data selection before training. This analysis does not require prohibitive computational costs due to finetuning as needed in the original paper.
>
> We will include this result in the revised version of the paper to clarify the practical applicability of our method to SFT data selection.
>
> **Diameter** (mean±var)
> | Dataset / Layer Ratio|0.1|0.3|0.5| 0.7 | 0.9 | MATH500 | AIME2024 |
> |---|---|---|---|---|---|---|---|
> | S1-v1.0|131.24±5.18| 360.36 ± 9.40 | 691.68 ± 11.52 | 1057.64±16.36 | 2006.44  ± 20.10| 92.6% |50.0% |
> | S1-v1.1| 140.54 ± 8.85| 392.33 ± 9.32| 783.55 ± 12.24 | 1100.68±17.07| 2036.03  ± 21.97|**94.4**% |**56.7**%|
>
> **>W4**
> > the reverse does not necessarily hold: SFT data with more cycles and larger diameters do not inherently lead to better performance.
>
> We agree with you that datasets with more cycles or larger graph diameters may not inherently guarantee better model performance. Our intention was not to suggest that **solely** optimizing for graph properties such as diameter will lead to better results.
>
> Rather, we position graph metrics such as diameter as auxiliary signals for data selection, not as standalone objectives. In practice, high-quality data pools should first be constructed using strong reasoning models (e.g., Gemini or DeepSeek-R1), as in the S1 pipeline. Once such data is available, reasoning graph metrics can serve as informative heuristics to filter or prioritize samples that are more likely to support effective reasoning. These metrics are meant to complement, not replace, other quality criteria.
>
> We explicitly do not advocate for selecting data solely to maximize graph diameter, as this could promote undesirable behaviors like overthinking. Instead, we propose reasoning graph analysis as a diagnostic tool, complementary to other strategies, to inform data selection.
>
> To make this intended usage clearer, we will revise the relevant sections in the paper.
>
> **>W5**
> > The attribution of redundant cycles to overthinking (Lines 296-298) conflicts with the empirical analysis in Section 4.4, which conclusively traces their origin to language-mixing effects.
>
> In Section 4.4, we presented language mixing as one concrete example that may lead to redundant cycles. Our mention of overthinking was not meant to contradict this, but rather to discuss another pattern of redundant cycles observed in prior work.
>
> During a rebuttal, we newly observed a failed output from Distill-Qwen-32B on GSM8K, which involved a simple arithmetic problem. Despite the simplicity, the model showed repetitive `Wait` outputs and trial-and-error loops, similar to patterns noted in [2]. This type of redundant cycle differs from the language-mixing discussed in Section 4.4 of the main paper.
>
> ```
> …
> They have 78 left, so S + 59 = 78 → S = 19.
> Same result. So, perhaps the problem is that the initial stock is 19, but they used 38 on Day 1, ...
> Wait, maybe the problem is that the orders are received before ...
> Wait, that's what I did initially. So, perhaps the problem is ....
> Wait, maybe I made a mistake in the equations. Let me try to model it again, ...
> Wait, perhaps the problem is that the orders are received before the usage on the...
> Wait, let me try that.
> …
> ```
>
> To avoid confusion, we will revise the relevant sentence to clarify that language mixing is one identified cause of redundant cycles, while overthinking represents another distinct pattern.
>
> [2] Do NOT Think That Much for 2+3=? On the Overthinking of o1-Like LLMs.
>
> **>W6**
> >The paper has many minor flaws that may affect the reader's impression:
>
> Thank you for pointing out the minor issues related to notation, citations, and descriptions. We appreciate your careful reading.
>
> We will thoroughly review and revise these details in the final version to improve clarity and presentation.
>
> **>Q3**
> >The authors should explain why they chose the hidden states from the layer of 90% depth as the default rather than the last layer.
>
> As there is no established consensus on which layer of an LLM best reflects reasoning-related representations, we conducted a comprehensive analysis using hidden states from five different depths: 10%, 30%, 50%, 70%, and 90%. The figures in the paper report results from all these layers. Among them, the 90% layer consistently showed the clearest trends, which is why we used it as the default in our main analysis.
>
> Following your suggestion, we also analyzed cycles using the hidden states from the final (**100%**) layer. As shown in the table below, the reasoning model (Distill-Qwen-32B) consistently exhibits more cycles than the base model (Qwen2.5-32B), and the final layer also follows the same trends. This result suggests that our analysis holds even at the final layer, supporting the robustness of our findings across different depths.
>
> **Cycles**
> | Model / Layer Ratio | 0.1  | 0.3  | 0.5  | 0.7  | 0.9  | 1.0  |
> |---|---|---|---|---|---|---|
> | Distill-Qwen-32B | 0.80 | 0.73 | 0.57 | 0.50 | 0.61 | **0.83** |
> | Qwen2.5-32B                   | 0.18 | 0.15 | 0.11 | 0.07 | 0.13 | **0.18** |
>
> **>Q4**
> >The authors could demonstrate the statistics (e.g., average number of nodes and edges)
>
> To address your question, we report below the average number of nodes and edges in the reasoning graphs generated for each dataset: GSM8K, MATH500, and AIME2024.
> The model used was Distill-Qwen-32B, with a target layer ratio of 0.9 and K = 200 for K-means clustering. As shown below, the number of nodes increases as task difficulty increases, reflecting the expansion of the reasoning search space.
> For a more detailed interpretation of what each node represents in the reasoning process, please refer to our response to **Reviewer mySX, W2**.
> We will include this table in the Appendix of the revised paper to improve the clarity.
>
> |Dataset| Avg. # Nodes | Avg. # Edges |
> |---|---|---|
> | GSM8K|18.66| 27.80|
> | MATH500 | 23.19| 60.44|
> | AIME2024 | 33.60 | 183.90 |

---

> > ### Comment · Reviewer_EMKy · 2025-08-04
> >
> > Thank you for your response. I have follow-up questions regarding the W2 and W3.
> >
> > The author may have misunderstood my point. In the S1 scenario, after collecting a large amount of SFT data (e.g., 59k samples), many low-quality samples are discarded, and only a small portion is retained (e.g., 1,000 samples). In such a case, how does the proposed method select high-quality samples (e.g., selecting 1,000 or another number from the 59k samples)? This reflects a more realistic application scenario, rather than the simplified setting described by the author where only two candidate subsets (e.g., S1-v1.0 and S1-v1.1) are available.
> >
> > In the response, the author mentioned that the graph diameter metric is not the only factor to consider when constructing SFT data. My question then is: in real-world scenarios, how can the graph diameter metric be effectively combined with other approaches to better select data samples?

---

> > > ### Author Response · Authors · 2025-08-05
> > >
> > > Thank you for the follow-up questions. We appreciate the opportunity to clarify the practical utility of our method in realistic SFT data selection scenarios.
> > >
> > > ---
> > >
> > > > In such a case, how does the proposed method select high-quality samples (e.g., selecting 1,000 or another number from the 59k samples)?
> > >
> > > To clarify a point that was already addressed in **W4**,  our proposed method is not intended to replace existing filtering procedures (such as those used in S1), but rather to function as a **lightweight auxiliary signal**. It is designed to be applied after a high-quality candidate pool has been constructed, typically using strong reasoning models such as Gemini or DeepSeek-R1. We believe the scenario described by the reviewer—selecting 1,000 samples from a pool of 59k—precisely corresponds to this filtering process. Again, our method is not meant to operate in isolation, but to be used as **one of the multiple signals that inform this selection.**
> > >
> > > Once a reasonably clean set of candidate samples is obtained, our graph-based metric (e.g., graph diameter) can serve as a useful heuristic for ranking or comparing samples. As demonstrated in W2&W3, even without any fine-tuning, high-quality SFT datasets (e.g., S1-v1.1) consistently induce larger reasoning graph diameters than low-quality ones (e.g., S1-v1.0). This suggests a **correlation between the potential reasoning capability enabled by an SFT dataset and the diameter of its induced reasoning graphs.**
> > >
> > > In the case of S1, various heuristics—such as quality filters, difficulty filters, and reasoning-length filters—are used to identify promising samples. Our experimental results indicate that the **graph diameter metric can serve as an additional, complementary indicator within this selection pipeline**.
> > >
> > > We will clarify these assumptions more explicitly in the revised version, to ensure there is no misunderstanding regarding the intended use and scope of our method.
> > >
> > > ---
> > >
> > > > how can the graph diameter metric be effectively combined with other approaches to better select data samples?
> > >
> > > As we noted in our rebuttal (**W4**), meaningless reasoning traces—such as those arising from language mixing or overthinking patterns like repetitive "Wait..." statements—can also increase graph diameter. Therefore, we emphasize that our method is most effective when **combined with complementary techniques** that target such pathological reasoning patterns.
> > >
> > > Fortunately, prior work provides tools to detect these failure cases:
> > >
> > > - For **language mixing**, recent work (e.g., [1]) has proposed robust language confusion metrics.
> > > - For **overthinking**, methods for identifying meaningless reasoning loops (e.g., [2]) are effective.
> > >
> > > We believe that combining these filtering methods with our graph-based analysis offers a practical and complementary framework for SFT data selection. In this framework:
> > >
> > > - Diameter serves as a proxy for the richness of reasoning.
> > > - Language confusion and meaningless loop detection help eliminate degenerate or pathological cases.
> > >
> > > By integrating these signals, practitioners can more reliably isolate high-quality reasoning traces in large, noisy datasets. We will make this intended usage clearer in the revised version of the paper.
> > >
> > > [1]  Marchisio, et al.  Understanding and Mitigating Language Confusion in LLMs. arXiv:2406.20052
> > > [2] Zhao, et al. Let LLMs Break Free from Overthinking via Self-Braking Tuning. arXiv:2505.14604

---

### Official Review · Reviewer_mySX · 2025-06-24

**Clarity:** 1
**Significance:** 2
**Originality:** 3
**Rating:** 4
**Confidence:** 3

**Summary:**

The paper analyzes the reasoning steps of CoT LLMs as graph, and find interesting correlations between graph properties and the reasoning capability of the LLMs.

**Questions:**

None

**Ethical Concerns:**

["NO or VERY MINOR ethics concerns only"]

**Final Justification:**

The authors responded to my concerns and the paper does have some novelty in its interesting perspective to view the reasoning LLM as a graph.

**Limitations:**

Yes

**Quality:**

2

**Strengths And Weaknesses:**

Strength:
1. This is a interesting perspective to view each type of reasoning steps as node and analyze the graph patterns between reasoning node.   The graph properties shows strong correlation to the reasoning capability of the Qwen model family.

Weakness:
1. The paper is a bit hard to read with some grammar issues. For example in the first paragraph "These recent reasoning models are characterized with to think and reason for longer before responding". Also the choice of math notations are also not helping the readers to better understand the paper. For example, the author used "T" to define the number of reasoning steps, then the readers will naturally expect  to use "t" to index the step, but then the author used "t" to index the token within each step. Same for "l" and "L". In general I find much can be improved in the writing.
2. For the clustering of segments, there are many ad hoc choices that are not explained well. For example, the choice of K=200. Also the authors only showed 3 representative centroids among the 200. Can the authors add the others in the Appendix? I am curious if the other centroid make sense.
3. It seems the experimental results are restricted to the Qwen family of models. The results will be stronger if more model families can be included.

---

> ### Author Rebuttal · Authors · 2025-07-31
>
> We sincerely thank the reviewers for their thoughtful and constructive feedback. We conducted additional analyses and clarifications to address your concerns, as detailed below.
>
> **>W1**
> >The paper is a bit hard to read with some grammar issues. For example in the first paragraph "These recent reasoning models are characterized with to think and reason for longer before responding". Also the choice of math notations are also not helping the readers to better understand the paper. For example, the author used "T" to define the number of reasoning steps, then the readers will naturally expect to use "t" to index the step, but then the author used "t" to index the token within each step. Same for "l" and "L". In general I find much can be improved in the writing.
>
> Thank you for pointing this out. Regarding the sentence
> `These recent reasoning models are characterized with to think and reason for longer before responding`
> we have corrected it to:
> `These recent reasoning models are characterized by their ability to think and reason for longer before responding`
>
> As for the notations, we agree that the original usage was potentially confusing. To improve clarity:
> - We now use $t$ to index the reasoning step,
> - and use $m$ to index the token within each step.
> - The layer index previously denoted by $\ell$ is now replaced by $\mu$.
>
> If there are any other specific points that you found difficult to follow or unclear, we would greatly appreciate further feedback so we can improve the paper’s readability.
>
> **>W2**
> > For the clustering of segments, there are many ad hoc choices that are not explained well. For example, the choice of K=200. Also the authors only showed 3 representative centroids among the 200. Can the authors add the others in the Appendix? I am curious if the other centroid make sense.
>
>
> As noted in lines 181–183 of the paper, we already included an analysis of different values of $k$ in **Appendix D**. Specifically, we tested $k = 50, 100$, and $200$, and found that the trends discussed in the main paper were consistent across these $k$-values. We invite you to refer to **Appendix D** for full details.
>
> Furthermore, to address your concern about the other centroids when $k = 200$, we conducted an automatic labeling experiment using an LLM to explore what kinds of reasoning patterns these centroids represent. Specifically, we used the `GPT-4o-mini` API to assign a theme to each centroid, based on the reasoning steps associated with it in the reasoning graph of DeepSeek-R1-Distill-Qwen-32B on the GSM8K dataset.
> We provided the following system prompt to GPT, and then input multiple reasoning steps corresponding to each centroid as the user prompt:
> ```
> You are a data analyst. The following is an output from a LLM.
> Your task is to carefully read the text and summarize its main theme in 1–3 English words.
> ```
>
> The following table presents the assigned **theme** for each centroid, along with **example** reasoning steps that were mapped to that centroid.
>
> | **Themes** |**Examples**  |
> |----|----|
> | **Calculations Totals (node 83)**     | 5 + 10 is 15. Then, 15 + 9 is 24. Finally, 24 + 3 is 27|
> |                             |Let me add them step by step. 500 + 1500 is 2000. Then, 2000 + 125 is 2125. |
> | **Calculations  Average (node 119)**    | \text{Average} = \frac{\text{Sum of all values}}{\text{Number of values}}
> |                              |\text{Average Speed} = \frac{\text{Total Distance}}{\text{Total Time}} = \frac{250 \, \text{miles}}{5 \, \text{hours}} = 50 \, \text{miles per hour}|
> | **Calculations Division (node 41)**   | \frac{125,000}{20} = 6,250|
> |                             | \frac{120 \, \text{pieces}}{15 \, \text{pieces/pack}} = 8 \, \text{packs}|
> |                             | \frac{80}{10} = 8 \text{ weeks}|
> | **Age Calculations (node 147)**          | Sum of their ages in two years = (B + 2) + (2B + 2) = 28|
> |                             |Brooke's age = B|
> |                             |C = 2 * (James’s age in 8 years) - 5|
> | **Cost Calculations (node 168)**       | $47.00 × 5 = $235.00|
> |                             |Keenan’s weekly cost = $160 ÷ 4 = $40|
> |                             |\( 8 \times 8 = \$64.00 \).|
> | **Answers Format (node 137)**        | The answer is: 50|
> |                             |The answer is: 48|
> |                             |The answer is: 8|
> | **Placeholder Tags (node 26)**        |<  /think>|
> |                             |< /think>|
> |                             |< /think>|
> | **Planning (node 163)**          | Okay, so I need to figure out how much Leah has spent on her new kitten so far. Let me break down the problem step by step.|
> |                             | Hmm, let's break it down step by step.|
> | **Calculation Correctness (node 52)** |Wait, that seems straightforward, but let me double-check…|
> |                              |Wait, let me double-check my calculations to make sure…|
> | **Reevaluation (node 100)**            |Wait, maybe I made a mistake in the equations.|
> |                             |Wait, maybe I made a mistake in the equations. Let me try to model it again,…|
> | **Inconsistencies (node 110)**         |Wait, that's a problem. 15 + 8 = 23, which is more than total time 20.|
> |                             |Wait, perhaps the shows are part of the 30%, and the other activities are part of the remaining 70%. But that doesn't make sense…|
>
> This analysis revealed that centroids often align with meaningful reasoning patterns. In addition to previously reported examples such as *Add*, *Multiply*, and *Wait*, we discovered centroids associated with higher-level computations (e.g., *Calculations Totals*, *Averages*), semantics-bound reasoning (e.g., *Age Calculations*, *Cost Calculations*), and structural elements (e.g., *Answer Formatting*, *Placeholder Tags*).
>
> Interestingly, we also identified centroids linked to reasoning behaviors, such as *Planning*, which reflects the model’s initial steps when approaching a math problem.
> Moreover, nodes like `Wait` appear to encompass diverse subtypes. For instance, we found centroids such as *Calculation Correctness* and *Reevaluation*, which reflects the model's tendency to reassess or double-check its own output.
> We also identified a distinct centroid labeled *Inconsistencies*, which captures moments where the model recognizes contradictions in its own reasoning and attempts to recalculate accordingly.
>
> **We believe that our additional analysis effectively addresses your concern regarding whether each centroid makes sense.**
>
> To improve clarity and better support the validity of our analysis, we will include the complete list of centroid themes with illustrative examples in the revised Appendix.
>
>
> **>W3**
> > It seems the experimental results are restricted to the Qwen family of models. The results will be stronger if more model families can be included.
>
> Recent studies on reasoning [1,2] have focused on the Qwen family. In line with this trend, we chose Qwen models for our main experiments due to their widespread use in recent reasoning research.
>
> To further test the generality of our findings, and following your suggestion, we conducted the same reasoning graph analysis using models from the LLaMA family. Specifically, we compared **LLaMA-3.1-8B (base model)** and **DeepSeek-R1-Distill-LLaMA-8B (reasoning model)** on the GSM8K dataset.
>
> The tables below present results for cycle ratio, graph diameter, and small world index. As with the Qwen models, we observe that the reasoning model exhibits more cycles, larger diameters, and larger small-worldness than the base model across all layer ratios. **This consistency supports the generality of our findings in another series of models.**
>
> We will include these results in the revised version of the paper. We will be happy to address if the reviewer has further suggestions on open-source reasoning and base model pairs, which need to be included.
>
> ---
> **Cycles**
> | Model / Layer Ratio                      | 0.1  | 0.3  | 0.5  | 0.7  | 0.9  |
> |----|----|----|----|----|----|
> | DeepSeek-R1-Distill-Llama-8B | 0.92 | 0.92 | 0.90 | 0.92 | 0.94 |
> | Llama-3.1-8B                  | 0.51 | 0.61 | 0.71 | 0.69 | 0.70 |
>
> ---
> **Diameter** (mean±var)
> | Model / Layer Ratio                          | 0.1        | 0.3       | 0.5   | 0.7 | 0.9  |
> |----|----|----|----|----|----|
> | DeepSeek-R1-Distill-Llama-8B   | 5.90 ± 1.37           | 26.68 ± 2.76          | 57.77 ± 4.31          | 105.87 ± 6.11         | 153.80 ± 7.56         |
> | Llama-3.1-8B                   | 1.67 ± 1.05           | 11.10 ± 2.29          | 19.35 ± 2.97          | 33.07 ± 3.93          | 46.80 ± 4.66          |
>
> ---
> **Small world index**
> | Model / Layer Ratio                              | 0.1    | 0.3  | 0.5    | 0.7    | 0.9     |
> |---------|----------|---------|----------|-----------|--------|
> | DeepSeek-R1-Distill-Llama-8B       | 0.70        | 0.73       | 0.72         | 0.73        | 0.78     |
> | Llama-3.1-8B                        | 0.24  | 0.36        | 0.44        | 0.45    | 0.47       |
>
> [1] Does Reinforcement Learning Really Incentivize Reasoning Capacity in LLMs Beyond the Base Model?
> [2] The Invisible Leash: Why RLVR May Not Escape Its Origin

---

> > ### Comment · Reviewer_mySX · 2025-08-05
> >
> > Thanks for the reply. This addressed the majority of my concerns and I decided to raise my score.

---

### Official Review · Reviewer_8fWp · 2025-06-30

**Clarity:** 2
**Significance:** 3
**Originality:** 3
**Rating:** 4
**Confidence:** 4

**Summary:**

The paper introduces the concept of reasoning graphs to analyze the internal mechanisms of large reasoning models. The paper systematically analyzes key graph-theoretic properties such as cyclicity, diameter, and small-world index across multiple tasks to provide insights into the success of reasoning models. The empirical findings suggest that large reasoning models exhibit unique structural properties in their reasoning graphs, which correlate with enhanced performance. The paper also reviews related works and discusses the implications of these findings for understanding large reasoning models and improving SFT-Data construction.

**Questions:**

1. Can you provide more details on the construction of reasoning graphs, especially regarding how the nodes are constructed and whether they are biased towards specific large reasoning models?
2. How do you justify the validity of the reasoning graphs in reflecting the true reasoning process of large reasoning models?
3. Can you elaborate on the implications for reasoning SFT-Data construction and how the proposed method can be used to directly judge the quality of SFT-Data?

**Ethical Concerns:**

["NO or VERY MINOR ethics concerns only"]

**Final Justification:**

I appreciate the interesting paradigm and findings presented in the paper, which could offer new insights and directions for the community. However, the proposed method relies on unresolved assumptions and appears to lack robustness in handling certain edge cases. Therefore, I maintain my initial score as weak accept.

**Quality:**

3

**Strengths And Weaknesses:**

## Paper Strength
1. Innovative Methodology: The introduction of reasoning graphs to analyze large reasoning models is innovative and provides a new perspective on understanding their internal mechanisms.
2. Insightful Empirical Findings: The paper offers insightful empirical findings on the structural properties of reasoning graphs in large reasoning models and how they correlate with enhanced performance.
3. Comprehensive Analysis: The paper systematically analyzes key graph-theoretic properties across multiple tasks, providing a comprehensive understanding of the reasoning processes in large reasoning models.

## Paper Weakness
1. Lack of Implementation Details: The construction of reasoning graphs needs further clarification. Are the nodes in the graphs constructed separately for each evaluated LLMs? or are they pre-constructed from the outputs of a single LLM such as Qwen2.5-32B-Instruct? If so, would it lead to a bias towards similar large reasoning models, as they tend to generate similar outputs? Besides, when evaluating the reasoning graphs in different layers, would the nodes in the graphs be constructed from the outputs of the same layer of the LLMs? These details are crucial for judging the validity of the findings in the paper.

2. Concerns regarding the reasoning graphs: Are the reasoning graphs really reflecting the true reasoning process of large reasoning models or just showing the statistics for characteristics of the output tokens? Because the reasoning step is broken down by empirically checking the `\n` and grouping the hidden states of output tokens. Each node in the reasoning graph is a composition of reasoning steps with similar semantics. **Would this really reflect the internal reasoning process of large reasoning models or just a statistical analysis of the output tokens?** For example, the paper mentions that the reasoning graphs exhibit cycles, which could be related to the repetition of the `wait` token in the LLMs outputs.

Considering the recent works [1] argue that the intermediate tokens in the reasoning process are just some statistically generated tokens and lack any real semantic content or algorithmic meaning (e.g., language mixing in DeepSeek-R1-zero). They could not be treated as reasoning traces of LLMs. This further raises my concerns about the validity of the reasoning graphs in the paper.

3. Implications for Reasoning SFT-Data Construction: The implications for reasoning SFT-Data construction are not well justified. The experiments in Section 5 only show that the model trained with the s1-v1.1 data can generate larger diameter reasoning graphs. This just reflects the reasoning patterns of the finetuned LLMs and is not a direct indication of the quality of the SFT-Data. Can we apply the proposed method to construct the reasoning graph offline and then analyze the graph characteristics to directly judge the quality of the SFT-Data? A naive solution I can think of is to feed the SFT-Data into the forward process of the LLMs in a teaching-forcing manner and extract the hidden states to construct the reasoning graph. By analyzing the cycles and the diameter of the reasoning graph, we can judge the quality of the SFT-Data to see if it is suitable for training the LLMs (i.e., more cycles and a larger diameter are better).

[1] Kambhampati, Subbarao, et al. "Stop Anthropomorphizing Intermediate Tokens as Reasoning/Thinking Traces!." arXiv preprint arXiv:2504.09762 (2025).

---

> ### Author Rebuttal · Authors · 2025-07-31
>
> Thank you for your thoughtful and constructive feedback. We carefully addressed each of your questions below and made several clarifications and extensions to strengthen the paper.
>
> **>W1&Q1**
> >Lack of Implementation Details
>
> >Can you provide more details on the construction of reasoning graphs, especially regarding how the nodes are constructed and whether they are biased towards specific large reasoning models?
>
> To clarify, the nodes in the reasoning graph are constructed independently for **each  model**. Therefore, there is no risk of bias arising from using pre-constructed nodes generated by a single model such as Qwen2.5-32B-Instruct. Each model produces its own reasoning steps, which are then segmented and embedded to form the graph.
>
> Similarly, when analyzing different layers, we construct segment representations separately for **each layer of each model**, as described in Lines 114–116 of the paper. Thus, the reasoning graph at each layer is based on the outputs from that specific layer.
>
> To improve clarity, we will revise the relevant sections in the paper to explicitly explain these details. We appreciate your feedback for helping us identify this point of confusion.
>
> **>W2&Q2**
> >Would this really reflect the internal reasoning process of large reasoning models or just a statistical analysis of the output tokens?
>
> >How do you justify the validity of the reasoning graphs in reflecting the true reasoning process of large reasoning models?
>
> To address this concern about the validity of reasoning graphs, we conducted an automatic labeling experiment using an LLM to examine whether the nodes (centroids) in the reasoning graph correspond to meaningful reasoning behaviors, rather than simply grouping surface-level token patterns.
> Specifically, we used the `GPT-4o-mini` API to assign semantic themes to each centroid in the reasoning graph of DeepSeek-R1-Distill-Qwen-32B, constructed from its reasoning traces on the GSM8K dataset. For each centroid, we provided the associated reasoning steps and prompted the model with the following instruction:
> ```
> You are a data analyst. The following is an output from a LLM.
> Your task is to carefully read the text and summarize its main theme in 1–3 English words.
> ```
>
> The following table presents the assigned theme for each centroid, along with example reasoning steps that were mapped to that centroid.
>
> This analysis revealed that centroids often align with meaningful reasoning patterns. In addition to previously reported examples such as Add, Multiply, and Wait, we discovered centroids associated with higher-level computations (e.g., *Calculations Totals*, *Averages, Divisions*), semantics-bound reasoning (e.g., *Age Calculations*, *Cost Calculations*), and structural elements (e.g., *Answer Formatting*, *Placeholder Tags*).
>
> Interestingly, we also identified centroids linked to reasoning behaviors, such as *Planning*, which reflects the model’s initial steps when approaching a math problem.
> Moreover, nodes like `Wait` appear to encompass diverse subtypes. For instance, we found centroids such as *Calculation Correctness* and *Reevaluation*, which reflect the model's tendency to reassess or double-check its own output.
> We also identified a distinct centroid labeled *Inconsistencies*, which captures moments where the model recognizes contradictions in its own reasoning and attempts to recalculate accordingly.
>
>
> | **Themes** |**Examples**  |
> |----|----|
> | **Calculations Totals (node 83)**     | 5 + 10 is 15. Then, 15 + 9 is 24. Finally, 24 + 3 is 27|
> |                             |Let me add them step by step. 500 + 1500 is 2000. Then, 2000 + 125 is 2125. |
> | **Calculations  Average (node 119)**    | \text{Average} = \frac{\text{Sum of all values}}{\text{Number of values}}
> |                              |\text{Average Speed} = \frac{\text{Total Distance}}{\text{Total Time}} = \frac{250 \, \text{miles}}{5 \, \text{hours}} = 50 \, \text{miles per hour}|
> | **Calculations Division (node 41)**   | \frac{125,000}{20} = 6,250|
> |                             | \frac{120 \, \text{pieces}}{15 \, \text{pieces/pack}} = 8 \, \text{packs}|
> |                             | \frac{80}{10} = 8 \text{ weeks}|
> | **Age Calculations (node 147)**          | Sum of their ages in two years = (B + 2) + (2B + 2) = 28|
> |                             |Brooke's age = B|
> |                             |C = 2 * (James’s age in 8 years) - 5|
> | **Cost Calculations (node 168)**       | $47.00 × 5 = $235.00|
> |                             |Keenan’s weekly cost = $160 ÷ 4 = $40|
> |                             |\( 8 \times 8 = \$64.00 \).|
> | **Answers Format (node 137)**        | The answer is: 50|
> |                             |The answer is: 48|
> |                             |The answer is: 8|
> | **Placeholder Tags (node 26)**        |<  /think>|
> |                             |< /think>|
> |                             |< /think>|
> | **Planning (node 163)**          | Okay, so I need to figure out how much Leah has spent on her new kitten so far. Let me break down the problem step by step.|
> |                             | Hmm, let's break it down step by step.|
> | **Calculation Correctness (node 52)** |Wait, that seems straightforward, but let me double-check…|
> |                              |Wait, let me double-check my calculations to make sure…|
> | **Reevaluation (node 100)**            |Wait, maybe I made a mistake in the equations.|
> |                             |Wait, maybe I made a mistake in the equations. Let me try to model it again,…|
> | **Inconsistencies (node 110)**         |Wait, that's a problem. 15 + 8 = 23, which is more than total time 20.|
> |                             |Wait, perhaps the shows are part of the 30%, and the other activities are part of the remaining 70%. But that doesn't make sense…|
>
> This analysis supports the claim that the reasoning graph provides a meaningful abstraction of the model’s reasoning process, rather than merely capturing surface-level token patterns. The presence of consistent and semantically interpretable themes across centroids—such as arithmetic operations, planning behaviors, and self-correction—indicates that the graph structure reflects underlying reasoning behaviors.
>
> To improve clarity, we will include the full list of labeled centroid themes and examples in the Appendix of the revised version.
>
>
> **>W3**
> > Can we apply the proposed method to construct the reasoning graph offline and then analyze the graph characteristics to directly judge the quality of the SFT-Data? A naive solution I can think of is to feed the SFT-Data into the forward process of the LLMs in a teaching-forcing manner and extract the hidden states to construct the reasoning graph.
>
> You are right in pointing out that the original version of our paper focused on analyzing reasoning graphs constructed post hoc—i.e., after training LLMs on different SFT datasets. While this analysis revealed interesting trends, it required training models before assessing the quality of the datasets.
>
> To improve its practicality for dataset construction with your suggestion, we extended our approach to evaluate reasoning graphs during the forward pass of a frozen model, without any retraining. Specifically, **we input various SFT datasets into a frozen LLM (Qwen2.5-32B-Instruct, the same model used in S1), and constructed reasoning graphs by extracting hidden states** using teacher forcing. This allowed us to evaluate the structural properties of the datasets prior to training.
>
> We applied this method to two datasets: S1-v1.0, and S1-v1.1, where it is known that models trained on S1-v1.1 achieve better downstream reasoning performance (e.g., higher accuracy on MATH500 and AIME2024). The table below compares the diameters of reasoning graphs induced by these datasets.
>
> We find that S1-v1.1 consistently yields larger graph diameters, even before any training has occurred, suggesting that reasoning graph properties can serve as a useful signal for evaluation of SFT data quality before training.
>
> Thank you again for the valuable suggestion. We will include this result in the revised version as a more direct method for evaluating SFT data quality.
>
> **Diameter** (mean±var)
> | Dataset / Layer Ratio     | 0.1       | 0.3       | 0.5   | 0.7     | 0.9    | MATH500 | AIME2024 |
> |----|----|----|----|----|----|----|----|
> | S1-v1.0           |131.24±5.18        |  360.36 ± 9.407   |   691.68 ± 11.52  | 1057.64±16.36    | 2006.44  ± 20.10   | 92.6% |50.0% |
> | S1-v1.1       | 140.54 ± 8.85         | 392.33 ± 9.32     | 783.55 ± 12.24       | 1100.68±17.07      | 2036.03  ± 21.97      | **94.4%** |  **56.7%**|

---

> > ### Comment · Reviewer_8fWp · 2025-08-01
> >
> > Thanks, authors, for the detailed response and interesting experiments. I have a remaining concern regarding this point:
> >
> > > Considering the recent works [1] argue that the intermediate tokens in the reasoning process are just some statistically generated tokens and lack any real semantic content or algorithmic meaning (e.g., language mixing in DeepSeek-R1-zero). They could not be treated as reasoning traces of LLMs. This further raises my concerns about the validity of the reasoning graphs in the paper.
> >
> > Although the proposed method groups reasoning steps using the hidden states of LLMs, it still extracts these steps by empirically detecting `\n` in the output tokens. Since some argue that tokens in the thinking process can be meaningless, is this approach a rigorous way to study the reasoning of LLMs? For example, DeepSeek-R1-Zero, trained from scratch with RL, sometimes mixes languages in its reasoning tokens. Despite their apparent meaninglessness, DeepSeek-R1-Zero performs well, demonstrating strong reasoning ability.
> >
> > Can the proposed extraction of reasoning steps and construction of reasoning graphs robustly handle such cases? Or does this approach assume that intermediate reasoning tokens are meaningful and directly reflect the reasoning of LLMs?
> >
> > I would appreciate further clarification on these points. If there are any assumptions required by the proposed method, including them in the paper would be helpful.

---

> > > ### Author Response · Authors · 2025-08-04
> > >
> > > We thank the reviewer for the thoughtful comments and for raising an important concern regarding the interpretation of intermediate tokens in the reasoning process.
> > >
> > > We find [1] to present an interesting perspective, and we agree with certain points, such as the idea of prompt argumentation—that generating more intermediate tokens can increase the likelihood of arriving at the correct answer. However, we understand that the view that intermediate tokens in reasoning models lack semantic meaning has **not** yet reached consensus in the community. For example, many studies [2, 3, 4], like ours, analyze intermediate tokens of reasoning models. In other words, multiple interpretations coexist, and efforts to clarify them empirically lie at the core of mechanistic interpretability research, as well as studies like ours that analyze the internal states of LLMs.
> > >
> > > Our central claim is that intermediate tokens in the reasoning process form graph structures with cycles and exhibit clearly different properties compared to those from base models. This observation holds regardless of whether the intermediate tokens carry real semantic content or not. In fact, we tried using prompts designed to encourage more structured outputs (i.e., methods that are more robust to language mixing), and we observed a drop in performance. This suggests that the cyclic graph structure itself may be key to improved reasoning performance.
> > >
> > > If one adopts the view that the intermediate tokens in the reasoning process do have semantic content, then our experiments in **W2&Q2** demonstrate that graph nodes can often be semantically mapped to individual reasoning steps. This provides meaningful insights into the model's internal processes. However, we emphasize that this is an empirical observation—we do not claim that “LLMs think like humans.” We also acknowledged that perfect mapping is not always possible, pointing to examples of language mixing and overthinking in our paper.
> > >
> > > On the other hand, even if one assumes that intermediate tokens in the reasoning process lack semantic content, our findings show that these tokens, compared to those from base models, exhibit more cycle structures and graph characteristics such as larger diameters. This provides a new angle of analysis.
> > >
> > > In any case, our analysis is entirely based on experimental observations and not influenced by any particular stance on whether reasoning models are genuinely "thinking." We hope it is understood that our work offers valuable new perspectives and insights—namely, that the intermediate tokens in reasoning processes form cyclic graph structures and exhibit distinct behaviors from those of base models.
> > >
> > > We plan to incorporate this discussion into the revised version of the paper.
> > >
> > >
> > > [1] Kambhampati, Subbarao, et al. "Stop Anthropomorphizing Intermediate Tokens as Reasoning/Thinking Traces!." arXiv preprint arXiv:2504.09762 (2025).
> > > [2] Kanishk, et al. "Cognitive Behaviors that Enable Self-Improving Reasoners,or,Four Habits of Highly Effective STaRs." arXiv preprint arXiv:2503.01307 (2025).
> > > [3] Constantin, Iv ́an, et al. "Understanding Reasoning in Thinking Language Models via Steering Vectors." arXiv preprint  arXiv:2506.18167 (2025).
> > > [4] Wang, et al. "Beyond the 80/20 Rule: High-Entropy Minority Tokens Drive Effective Reinforcement Learning for LLM Reasoning." arXiv preprint  arXiv:2506.01939 (2025).

---

### Official Review · Reviewer_dTpT · 2025-07-03

**Clarity:** 3
**Significance:** 3
**Originality:** 3
**Rating:** 5
**Confidence:** 5

**Summary:**

This paper introduces an interpretability method that extracts reasoning graphs from reasoning traces generated by large language models when solving complex tasks. These graphs are created by clustering the hidden state representations of models at each intermediate reasoning step when solving problems across an evaluation set. The centroids of these clusters are nodes, and transitions between reasoning steps for a given solution define the directed edges in the graph. The authors compare graph-theoretic properties of graphs constructed from traces taken from reasoning models and their non-reasoning, “base” counterparts to show how these metrics correlate with reasoning ability. They claim that this method illuminates how reasoning models approach complex problems, and they argue that these metrics can be used to build better reasoning datasets.

**Questions:**

(See questions above)

1. Figure 2 lists a small set of nodes taken from the reasoning graph as well as representative example reasoning steps for each node. The three nodes included are labelled “multiplicative,” “additive,” and “wait.” Are the nodes in the reasoning graphs studied in the paper always directly tied to concrete reasoning techniques? If we set K=200, then are all 200 nodes clearly associated with these techniques?

**Ethical Concerns:**

["NO or VERY MINOR ethics concerns only"]

**Final Justification:**

The authors addressed my concerns and questions - I am sticking with my original accept rating.  The paper adds a novel contribution to this field, and I believe their methodology and experiments are quite thorough.

**Limitations:**

yes

**Quality:**

3

**Strengths And Weaknesses:**

Strengths:
* The authors test their method across a wide range of model sizes and evaluation datasets. They sweep across different hyperparameters like k-values for clustering and the layer at which hidden state representations are extracted, showing that relationships presented are not artifacts of cherry picked graph construction. Additionally, the authors demonstrate how these metrics change at different depths within the model.
* The experiments are comprehensive and thoroughly demonstrate how different graph properties correlate with reasoning performance at different model sizes and target layers.  The examination of the nodes in the graphs, along with example reasoning steps, was very helpful for understanding what the reasoning graph might look like.
* One question while reading the paper is whether increases in these metrics are merely correlated with superior reasoning or whether reasoning models actually learn to leverage relevant techniques that are measured by these graph properties. The intervention study of training on s1 and iteratively measuring diameter distribution helps answer this question. Additionally, the comparison between s1 v1 and s1 v1.1 demonstrates that graph metrics not only increase with supervised finetuning on reasoning data but that they increase more on better data, supporting the authors’ claim that this method can be used for dataset curation.

Weaknesses:
* One possible confound is that these graph metrics may increase with longer reasoning chains in general. An experiment controlling for token length/flops is needed.
* The reasoning steps are segmented at newlines. Some models may have a propensity to generate longer sequences. Is comparing the graphs generated from these models fair?
* Figures 7b and 7c show a massive increase in AIME24 accuracy between Distill Qwen-7B and Qwen-14B. However, the cycle counts are similar between the two, and graph diameter is noticeably larger for Qwen-7B. Is there any explanation for why this massive increase in accuracy is not correlated with these metrics?
* Figure 3 demonstrates how DeepSeek-R1-Distill Qwen 32B exhibits more exploration and cyclicity in its reasoning graph compared to the corresponding base model. Were the examples presented in this figure solved by both models? If not, are all the nodes explored by the reasoning model essential to solving the answer. Recent papers like [1] claim that reinforcement learning with verifiable rewards improves pass@k at small k but not at large k, suggesting that RLVR does not expand the coverage of the base model. Do you think that overlaying graphs from multiple attempts from a base model would results in similar graph coverage to that of the reasoning model? Would studying the trajectory of the reasoning model in these problems help confirm or disprove the claim above?
* Additional insight into what the cycles accomplish would be helpful. The discussion around figure 7a claims that cyclic reasoning generally enhances reasoning effectiveness except in cases like language mixing. The authors later state in section 6 that “redundant cyclic structures may explain overthinking.” Is there any way to differentiate between trivial and non-trivial cycles that harm or help reasoning performance? In addition, if the nodes are tied to general math operations as suggested in figure 2, would something relatively simple like adding and multiplying numbers in sequence manifest as a cycle?


[1] [Does Reinforcement Learning Really Incentivize Reasoning Capacity in LLMs Beyond the Base Model?
](https://arxiv.org/abs/2504.13837)

---

> ### Author Rebuttal · Authors · 2025-07-31
>
> We thank the reviewer for their thoughtful comments and valuable suggestions, which helped us strengthen our analysis and clarify key aspects of our work.
>
> **>W1&W2**
>
> >One possible confound is that these graph metrics may increase with longer reasoning chains in general.
>
> > The reasoning steps are segmented at newlines. Some models may have a propensity to generate longer sequences. Is comparing the graphs generated from these models fair?
>
> First, we note that some of the key graph metrics we use—such as the small-world coefficient—are normalized by definition (line 154). Specifically, small-worldness is computed by dividing the graph’s clustering coefficient and average path length by those of a random graph with the same number of nodes and edges. This normalization is designed to enable fair comparisons even when graphs vary in size, such as when models differ in the number of reasoning segments.
>
> Second, our comparisons are not limited to base vs. reasoning models. We also analyze multiple reasoning models of different parameter sizes, all of which tend to produce long outputs. In those comparisons as well, we observe consistent trends: models with stronger reasoning performance tend to exhibit more structured graph properties, such as more cycles and higher small-worldness. This supports the claim that the graph metrics reflect reasoning capability rather than just output length or compute.
>
> Finally, and most directly addressing your suggestion, we did attempt experiments to control for output length by engineering prompts that elicit structured, similar-length responses across models. However, we found that **constraining the output in this way significantly degraded the performance of the reasoning models**. Therefore, in the main paper, we report results using unconstrained prompts, which better reflect the models’ natural and effective operating regimes.
>
> For these reasons, we believe the graph properties we observe reflect real differences in how the models reason, not just differences in output length or how much they compute.
>
> **>W3**
> >Figures 7b and 7c show a massive increase in AIME24 accuracy between Distill Qwen-7B and Qwen-14B. However, the cycle counts are similar between the two, and graph diameter is noticeably larger for Qwen-7B.
>
> Figures 7 only present the cycle ratio, cycle count, and diameter metrics, but **Appendix G** includes results for the small-world index, which shows that the 14B model exhibits higher small-worldness than the 7B model.
> The table below summarizes the average values of all four graph properties—cycle ratio, cycle count, diameter, and small-world index—based on the figures in the paper. As shown, the 14B model demonstrates better graph properties than the 7B model on **three out of four** metrics．
>
> This suggests that, when considering the full set of graph metrics, there is a meaningful relationship between stronger graph properties and better model performance.
> | Metric / Model-size      | 7B  | 14B  |
> |---|---|---|
> | Cycles | 0.93 | **1.00** |
> | Cycle counts | 3.56 |**4.10**|
> | Diameter | **2215.39** |1674.35 |
> | Small world index| 0.85 |**0.95** |
>
> **>Q1**
> >Are the nodes in the reasoning graphs studied in the paper always directly tied to concrete reasoning techniques?
>
> To answer your question, we conducted an automatic labeling experiment using an LLM to investigate whether the nodes in the reasoning graph correspond to meaningful reasoning techniques.
>
> Specifically, we used the `GPT-4o-mini` API to assign a theme to each node, based on the reasoning steps associated with nodes in the reasoning graph of Distill-Qwen-32B on the GSM8K dataset.
> We provided the following system prompt to GPT, and then input multiple reasoning steps corresponding to each node as the user prompt.
> ```
> You are a data analyst. The following is an output from a LLM.
> Your task is to carefully read the text and summarize its main theme in 1–3 English words.
> ```
> The table below presents some of the identified themes along with reasoning steps corresponding to each node. In addition to previously reported categories like Add, Multiply, and Wait, we discovered centroids associated with:
> - Higher-level computations (Calculations Totals, Averages, Divisions)
> - Semantics-bound reasoning (Age Calculations, Cost Calculations)
> - Structural elements (Answer Formatting, Placeholder Tags)
> - Reasoning behaviors such as Planning
> - Variants of "Wait" behavior, including Reevaluation, Inconsistencies, and Calculation Correctness
>
> | **Themes** |**Examples**  |
> |---|---|
> | **Calculations Totals (node 83)**     | 5 + 10 is 15. Then, 15 + 9 is 24. Finally, 24 + 3 is 27|
> | |Let me add them step by step. 500 + 1500 is 2000. Then, 2000 + 125 is 2125. |
> | **Calculations  Average (node 119)**    | \text{Average} = \frac{\text{Sum of all values}}{\text{Number of values}}
> |  |\text{Average Speed} = \frac{\text{Total Distance}}{\text{Total Time}} = \frac{250 \, \text{miles}}{5 \, \text{hours}} = 50 \, \text{miles per hour}|
> | **Calculations Division (node 41)**   | \frac{125,000}{20} = 6,250|
> |  | \frac{120 \, \text{pieces}}{15 \, \text{pieces/pack}} = 8 \, \text{packs}|
> | | \frac{80}{10} = 8 \text{ weeks}|
> | **Age Calculations (node 147)**  | Sum of their ages in two years = (B + 2) + (2B + 2) = 28|
> ||Brooke's age = B|
> |  |C = 2 * (James’s age in 8 years) - 5|
> | **Cost Calculations (node 168)**       | $47.00 × 5 = $235.00|
> | |Keenan’s weekly cost = $160 ÷ 4 = $40|
> | |\( 8 \times 8 = \$64.00 \).|
> | **Answers Format (node 137)** | The answer is: 50|
> |  |The answer is: 48|
> |  |The answer is: 8|
> | **Placeholder Tags (node 26)**|<  /think>|
> |   |< /think>|
> | |< /think>|
> | **Planning (node 163)**| Okay, so I need to figure out how much Leah has spent on her new kitten so far. Let me break down the problem step by step.|
> | | Hmm, let's break it down step by step.|
> | **Calculation Correctness (node 52)** |Wait, that seems straightforward, but let me double-check…|
> ||Wait, let me double-check my calculations to make sure…|
> | **Reevaluation (node 100)**|Wait, maybe I made a mistake in the equations.|
> | |Wait, maybe I made a mistake in the equations. Let me try to model it again,…|
> | **Inconsistencies (node 110)** |Wait, that's a problem. 15 + 8 = 23, which is more than total time 20.|
> |  |Wait, perhaps the shows are part of the 30%, and the other activities are part of the remaining 70%. But that doesn't make sense…|
>
> These findings suggest that many nodes align with concrete reasoning techniques, supporting the validity of the reasoning graph.
>
> While not all 200 nodes are directly tied to concrete reasoning operations—some, like `</think>`, reflect structural or formatting artifacts commonly produced by the model—the majority of nodes correspond to interpretable reasoning patterns.  This demonstrates that the reasoning graph captures valid reasoning techniques. We believe this supports the soundness of our approach.
>
> To improve clarity, we will include the full list of labeled node themes and examples in the Appendix of the revised version.
>
> **>W5**
> > Additional insight into what the cycles accomplish would be helpful.
>
> As you point out, not all cycles may contribute equally to reasoning quality, and differentiating between helpful and redundant cycles is a key challenge.
> To provide further insight, we conducted an additional analysis comparing the reasoning graphs of samples that were answered **correctly** (n = 936) and **incorrectly** (n = 64) by the reasoning model (Distill-Qwen-32B). The results are summarized below:
>
> | Metric | Correct | Incorrect |
> |----|----|----|
> | Cycle Ratio | 0.67 | 0.92 |
> | Diameter (mean ± var) | 5970.50 ± 45.10 | 3666.34 ± 34.93 |
>
> Interestingly, incorrect samples exhibit a higher proportion of cycles, yet have smaller graph diameters. This suggests that not all cycles are beneficial—specifically, cycles that do not significantly increase the diameter may represent shallow or localized loops that fail to expand the reasoning space meaningfully.
>
> This observation supports our interpretation in the main paper (e.g., discussion around language mixing and overthinking) that redundant cyclic structures may harm performance, and only cycles that meaningfully contribute to graph expansion (i.e., diameter) are potentially beneficial.
>
> This highlights the difference between trivial and non-trivial cycles and suggests new directions for reasoning graph analysis.
>
>
> **>W4**
> > Were the examples presented in this figure solved by both models? If not, are all the nodes explored by the reasoning model essential to solving the answer.
>
> > Do you think that overlaying graphs from multiple attempts from a base model would results in similar graph coverage to that of the reasoning model?
>
>
> As discussed in our response to **W5**, we found that cyclic patterns that do not contribute to increased diameter tend to be less helpful for reaching the correct answer.
>
> You also raise an interesting hypothesis: whether aggregating reasoning graphs from multiple base model attempts (e.g., pass@k) could resemble the graph produced by a single pass of a reasoning model. While this is a valuable line of inquiry, we emphasize a key distinction:
> In our setting, the reasoning graph is built from a single, coherent forward pass, where all reasoning steps are connected. We quantified the coverage of the graph by measuring diameter among the connected graph nodes. By contrast, pass@k consists of multiple disjoint generations, which may not form a connected graph under our definition of diameter.
> As a result, even if the node coverage is similar between the aggregated pass@k graph and the reasoning model's graph, their diameters need not align, since disconnected components are excluded from the diameter computation.
>
> However, we think this is a promising research direction, and exploring pass@k phenomena through the lens of graph theory could yield deeper insights into model exploration.

---

> > ### Comment · Reviewer_dTpT · 2025-08-03
> > **Acknowledge of response**
> >
> > I thank the authors for their response.  I have no further questions, and am satisfied with the quality of their answers to my review questions.  I will continue to advocate for the acceptance of this submission.

---

### Note · Authors · 2025-08-13

Dear Reviewers, AC, and SAC,

We would like to thank you for your great effort in reviewing our paper.
We believe our author responses have addressed all the raised concerns.
 Again, we would like to highlight some important discussions and follow-up experiments as follows. It would be great if you take these into consideration for your final decision.

**1) Content captured by reasoning-graph nodes** (Reviwers **dTpT, 8fWp, mySX**)

Building on the analysis in Fig. 2(b), we extended our study during the discussion phase to examine which reasoning steps each node represents. Specifically, we conducted an automatic centroid-labeling experiment using the GPT-4o API.
The induced themes include arithmetic operations (Totals, Averages, Divisions), semantics-bound skills (Age/Cost calculations), structural elements (Answer formatting, placeholder tags), and other reasoning behaviors (Planning, Calculation-correctness, Reevaluation, Inconsistency detection). This supports that many nodes align with concrete reasoning behaviors rather than superficial token statistics, strengthening the validity of our analysis and claims.

**2) Practical guidance for SFT data selection** (Reviwers **8fWp, EMKy**)

To extend the Section 5 setting toward practical SFT construction, we evaluated candidate SFT datasets via teacher forcing on a frozen model (Qwen2.5-32B-Instruct). Without any finetuning, higher-quality data (S1-v1.1) consistently induces larger reasoning-graph diameters than S1-v1.0, mirroring its downstream gains. These results further support our claim that the reasoning-graph diameter can serve as an auxiliary signal for selecting and constructing higher-quality SFT data.

**3) Generality across models and tasks** (Reviwers **mySX, EMKy**)

While the main paper analyzed the Qwen family, we extended the experiments to the LLaMA family as well. We also expanded beyond the three math tasks (GSM8K, MATH500, AIME2024) to include StrategyQA and Logical Deduction. In all cases, reasoning models show more cycles, larger diameters, and higher small-worldness. These findings reinforce the robustness of our results across models and tasks.

Thank you again for your time and for the many suggestions that improved the paper.

---

### Decision · Program_Chairs · 2025-09-17

**Decision:**

Accept (poster)

**Comment:**

This paper offers a novel contribution on how large language models reason by leveraging reasoning graphs. Its extensive empirical evaluation and methodology are strengths that have been praised by reviewers. However, concerns about whether the graph metrics merely reflect generic properties (such as sequence length) rather than true reasoning, along with questions about the generalizability and practical applicability of the findings, have been raised in the reviews as well as some clarity and presentation issues. The authors (props to them!) managed to address the most important points imho.

Overall the aggregated reviews lean toward acceptance with some reviewers giving accept while others a borderline accept due to the  questions above. I think the paper is ready to be accepted.